# Mitigating over-exploration in latent space optimization using LES

**Omer Ronen** [1]  **Ahmed Imtiaz Humayun** [2]  **Richard Baraniuk** [2]  **Randall Balestriero** [3]  **Bin Yu** [1]

## Abstract

We develop Latent Exploration Score (LES) to mitigate over-exploration in Latent Space Optimization (LSO), a popular method for solving black-box discrete optimization problems. LSO utilizes continuous optimization within the latent space of a Variational Autoencoder (VAE) and is known to be susceptible to over-exploration, which manifests in unrealistic solutions that reduce its practicality. LES leverages the trained decoder's approximation of the data distribution, and can be employed with any VAE decoder–including pretrained ones–without additional training, architectural changes or access to the training data. Our evaluation across five LSO benchmark tasks and twenty-two VAE models demonstrates that LES always enhances the quality of the solutions while maintaining high objective values, leading to improvements over existing solutions in most cases. We believe that new avenues to LSO will be opened by LES' ability to identify out of distribution areas, differentiability, and computational tractability.

## 1. Introduction

Many important tasks in scientific fields, such as small molecule discovery and protein engineering, can be framed as discrete black-box optimization problems. In contrast to conditional sampling-based approaches, including GFlowNet (Bengio et al., 2023) and Diffusion (Corso et al., 2022; Igashov et al., 2024), which are better suited for applications like linker design (Du et al., 2024), optimization is particularly effective when the goal is to improve a specific property, such as enhancing a drug's safety.

LSO was recently developed to enhance the sample efficiency of discrete optimization algorithms, such as genetic algorithms, in the black-box setting (Gómez-Bombarelli

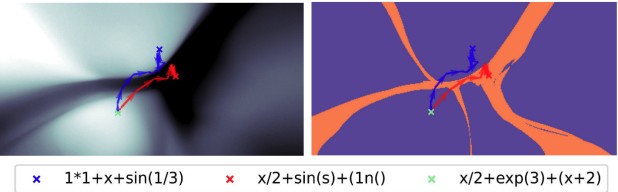

*Figure 1.* Incorporating LES promotes valid solutions. We consider the task of approximating the expression `1/3 + x + sin(x * x)`, using LSO. Optimization trajectories with (blue) and without (red) LES constraint in the latent space of a VAE are projected onto a two-dimensional subspace that contains the starting point and the end-points obtained after 10 gradient ascent steps. In the left panel, we show the LES score for latent vectors on the two-dimensional subspace, with darker shades corresponding to lower LES. In the right panel, we show the validity of the decoder outputs for each latent vector, with orange denoting invalid generations. High LES values correlate with valid areas, and incorporating LES in LSO produce an expression that adheres to the grammatical rules of Example 1.1.

et al., 2018). LSO transfers the optimization problem to the domain of the latent space of a VAE, which can be efficiently explored using continuous optimization methods. However, ensuring that LSO solutions respect the structure of the original space remains a challenge. To illustrate this issue, we provide some examples.

**Example 1.1** (Arithmetic expressions)**.** *An expression built up using numbers, arithmetic operators and parentheses is called an arithmetic expression. However, not every sequence of the above elements correspond to a valid expression. For instance, the expression* `"sin(x) + x"` *is a valid expression while* `"sin(xxx"` *is not.*

**Example 1.2** (Simplified molecular-input line-entry system (SMILES))**.** *SMILES provides a syntax to describe molecules using short ASCII strings. Atoms are represented by letters (e.g., water:"O"), bonds are represented by symbols (e.g., triple: "#", double: "=", . . .), branches are represented in parentheses and cyclic structures are represented by inserting numbers at the beginning and the end. Like the arithmetic expressions case, not every combination of the elements described above corresponds to a valid molecule. For example, while* `"C1CCCCC1"` *is valid, both* `"C1CCCCC2"` *and* `"C1CCCCC)"` *are not.*

[1]UC Berkeley [2]Rice Univeristy [3]Brown University. Correspondence to: Omer Ronen <omer_ronen@berkeley.edu>.

*Proceedings of the 42nd International Conference on Machine Learning*, Vancouver, Canada. PMLR 267, 2025. Copyright 2025 by the author(s).

**Example 1.3** (Quality filters for molecules)**.** *Chemists seek molecules that not only optimize desired chemical properties but are also stable and easy to synthesize. This has led to the development of rules such as Lipinski's Rule of Five (RO5 Lipinski et al. (1997)), which helps determine if the bioavailability (i.e., the proportion of a drug or other substance that enters the circulation when introduced into the body) of a given compound meets a certain threshold. For example, RO5 suggests that poor absorption is more likely when the octanol-water partition coefficient (logP) exceeds 5. Similarly, the Pan Assay Interference Compounds (Baell & Holloway, 2010) filter helps in identifying false positives in assay screenings. The* `rd_filters` *(Walters, 2019) package has curated many such rules and is considered a "high precision, low recall surrogate measure" (Brown et al., 2019). Following (Notin et al., 2021) we consider a sample valid if it passes the* `rd_filters` *filters* [1]

Numerous directions have been explored to overcome the challenge of providing valid solutions, including specialized VAE architectures (Kusner et al., 2017; Jin et al., 2018) or robust representations for discrete data (Krenn et al., 2020). Additionally, constrained objectives can be formulated under the assumption that one has access to a function which quantifies the validity of any point in the latent space (Griffiths & Hernández-Lobato, 2020). However, in many realistic scenarios, such as Example 1.3, these solutions may not be directly applicable, as the structure of the sequence space may not be sufficiently well understood. To address this, (Notin et al., 2021) proposed using an estimator of the uncertainty of the decoder, based on the variational approximation to a posterior distribution over the VAE parameters, encouraging LSO to respect the sequence space structure (details are provided Appendix D). Although this approach proved effective, the non-differentiable nature of the uncertainty score required its integration into LSO through heuristic approaches. Additionally, the computation of the uncertainty score is not exact (i.e., it relies on variational approximation and Monte Carlo sampling) and requires significant amount of time to compute. Therefore, there is a need for robust methods that work across different VAEs and sequence space structures, and can be easily integrated into existing LSO pipelines.

To achieve this goal, we develop LES, a score that can be used as a constraint in LSO optimization to increase the number of solutions that respect a given structure. The distinctive characteristics of LES are differentiability and robustness that allow its easy integration into existing LSO pipelines. Specifically, our contributions are as follows:

- We introduce LES, a score that achieves higher values in regions of the latent space closer to the training data.

---

[1] We use the default Inpharmatica rule set comprised of 91 alerts

Our results demonstrate that LES is highly effective at identifying regions that preserve the structure of the sequence space. Although LES' computation scales cubically with the latent dimension, it is up to 80% faster than the current state-of-the-art for identifying out-of-distribution data points in the latent space of generative models for discrete sequences (Tables 1 and 22).

- We develop a numerically stable optimization procedure to incorporate LES as a constraint in LSO.

- We evaluate LES-constrained LSO across thirty optimization tasks, including twenty-two VAEs and five benchmark problems, demonstrating its robustness in generating valid solutions and achieving high objective values. Specifically, in 19 out of the 30 LSO experiments, our method either finds the best solution on average or achieves a solution within 1 standard deviation of the best solution across 10 independent runs. This outperforms the six alternative methods we considered by 19% (Tables 20 and 21).

## 2. Background: Latent Space Optimization

LSO is a method for solving black box optimization problems in discrete and structured spaces, such as the space of valid arithmetic expressions. Formally, let $\mathcal{V} \subset \mathbb{R}^{L \times D}$ be a discrete and structured space, represented as a sequence of L one-hot vectors of dimension D. We represent sequences of length L of categorical variables with D categories. L is set as the maximum sequence length that we are optimizing for, and one of the D categories is used as an "empty" category (which enables generating sequences shorter than the maximal length). For instance, in the case of valid arithmetic expressions, $\mathcal{V}$ would be the set of sequences that define such expressions. Let $\mathcal{M} : \mathcal{V} \to \mathbb{R}$ be the objective function. LSO solves,

$$\arg\max_{x \in \mathcal{V}} \mathcal{M}(x). \tag{1}$$

In this setting, we assume that evaluations of the objective function ($\mathcal{M}$) are expensive to conduct. For example, the objective may be the binding affinity of a compound to a given protein, measured through a wet lab experiment.

A popular approach to solve Equation (1) is Bayesian Optimization (BO), which utilizes first order optimization of a surrogate model for $\mathcal{M}$. However, since the space is discrete, first order optimization cannot be directly applied. In an attempt to make BO applicable for solving Equation (1), (Gómez-Bombarelli et al., 2018) proposed to transfer the optimization problem into that over a domain of the latent space of a deep generative model and subsequently perform BO in this space. The main idea is to (1) learn a continuous representation of the discrete objects (e.g., using a VAE) and (2) perform BO in the latent space while

decoding the solution at each step. Formally, given a pre-trained encoder ($\mathbf{E}_\theta$) and decoder ($\mathbf{G}_\theta$) the initial labelled dataset $\mathcal{D} = \{\boldsymbol{x}_i, y_i\}_{i=1}^n$ is first encoded into the latent space $\mathcal{D}^{\boldsymbol{z}} = \{\boldsymbol{z}_i = \mathbf{E}_\theta(\boldsymbol{x}_i), y_i\}_{i=1}^n$. Using the encoded dataset, a BO procedure is conducted, which we describe in Algorithm 1. Most commonly, a Gaussian process is used as the surrogate model for $\mathcal{M}$, and the acquisition function is the expected improvement (Frazier, 2018), defined as:

$$\mathcal{A}_{\hat{f}}(\boldsymbol{z}) = \mathbb{E}_{\hat{f}} \max(\hat{f}(\boldsymbol{z}) - \max_i y_i, 0) \qquad (2)$$

where the expectation is with respect to the distribution of the function $\hat{f}$, conditioned on $\mathcal{D}^{\boldsymbol{z}}$.

---

**Algorithm 1** Latent Space Optimization

**for** $t = 1$ **to** $T$ **do**

1. Fit a surrogate model $\hat{f}$ to the encoded dataset, $\mathcal{D}^{\boldsymbol{z}}$

2. Generate a new batch of query points by optimizing a chosen acquisition function ($\mathcal{A}$)

$$\boldsymbol{z}^{(\text{new})} = \arg\max_{\boldsymbol{z}} \mathcal{A}_{\hat{f}}(\boldsymbol{z}) \qquad (3)$$

3. Decode $\boldsymbol{x}^{(\text{new})} = \mathbf{G}_\theta(\boldsymbol{z}^{(\text{new})})$, evaluate the corresponding true objective values ($y^{(\text{new})} = \mathcal{M}(\boldsymbol{x}^{(\text{new})})$) and update $\mathcal{D}^{\boldsymbol{z}}$ with ($\boldsymbol{z}^{(\text{new})}, y^{(\text{new})}$).

---

**Over-exploration in LSO**   Multiple studies (Notin et al., 2021; Kusner et al., 2017) have found that unconstrained latent space optimization (LSO) often yields solutions that disregard the aforementioned structures. For example, when searching for arithmetic expressions, invalid equations like "$ssin(xxx$" frequently occur. Similarly, many solutions in molecule searches fail to pass basic quality filters (Example 1.3), limiting their practical utility (Maus et al., 2022).

While acquisition functions such as expected improvement (Equation (2)) are designed to balance exploration and exploitation based on the estimated uncertainty from the Gaussian process model for $\mathcal{M}$. The frequent generation of invalid solutions during acquisition optimization, which implies that the estimated uncertainty can be problematic in this setting, underscores the need for additional regularization (Tripp et al., 2020), which we aim to address.

To mitigate over-exploration, we propose adding a penalty to Equation (3). The penalty uses a new score, giving higher values over the latent space valid set, defined as:

$$\{\boldsymbol{z}; \mathbf{G}_\theta(\boldsymbol{z}) \in \mathcal{V}\}, \qquad (4)$$

where $\mathbf{G}_\theta : \mathcal{Z} \to \mathbb{R}^{\text{L} \times \text{D}}$ is the decoder network, and $\mathcal{V} \subset \mathbb{R}^{\text{L} \times \text{D}}$ is the set of valid sequences.

The derivation of our score leverages the Continuous Piecewise Affine (CPA) representation of neural networks, which we briefly review below.

**Deep generative networks as CPA**   Following (Humayun et al., 2022; 2021; Balestriero & Baraniuk, 2018; Balestriero et al., 2024), we consider the representation of Deep Generative Networks (DGNs) as Continues Piecewise Affine (CPA) Splines operators. Let $f_\theta$ be any neural network with affine layers and piecewise affine activations then it holds that

$$f_\theta(\boldsymbol{z}) = \sum_{\omega \in \Omega} \left(\boldsymbol{A}_\omega \boldsymbol{z} + \boldsymbol{b}_\omega\right) 1_{\{\boldsymbol{z} \in \omega\}}, \qquad (5)$$

where $\Omega$ is the input space partition induced by $f_\theta$, $\omega$ is a particular region and the parameters $\boldsymbol{A}_\omega$ and $\boldsymbol{b}_\omega$ defines the affine transformation depending on $\omega$.

In cases where the neural network $f_\theta$ is not composed solely of piecewise affine layers and activations, we leverage the result from (Daubechies et al., 2022) to assert that Equation (5) either exactly represents $f_\theta$ or provides a sufficiently accurate approximation for our practical purposes (Humayun et al., 2022). We therefore argue that all the decoder neural networks included in our study (i.e., GRU, LSTM, and Transformers) can be approximated with high accuracy as continuous piecewise affine (CPA) functions.

## 3. A Latent Exploration Score to Reduce Over-Exploration in LSO

In this section, we introduce Latent Exploration Score (LES), our new score to reduce over-exploration in LSO. We begin by motivating LES and proceed to formally derive it in Section 3.1. In Section 3.2, we provide empirical evidence that LES gives higher values in the latent space valid set. The use of LES to regularize LSO is left for Section 4.

**Motivation**   Our goal is to develop a meaningful constraint for optimizing the acquisition function within a latent space of a given VAE. Specifically, we aim to construct a constraint that is a continuous function of $\boldsymbol{z}$ with higher values, indicating that it is more likely that $\boldsymbol{z}$ resides within valid regions of the latent space (Equation (4)).

Such a score should be higher in regions near training data points, assuming most of VAE training data is valid. To achieve this, we treat the latent space of the VAE as a probability space, i.e. $\boldsymbol{z} \sim p_{\boldsymbol{z}}$, for some prior distribution $p$ (for example standard Gaussian). The prior should reflect our best guess for the distribution of the observed data in the latent space. Solutions are mapped back to sequences by the decoder through a deterministic (we do not consider $\boldsymbol{x}$ to follow a conditional distribution given $\boldsymbol{z}$) transformation of the latent vectors. Therefore, any distribution on the latent

space defines a distribution over the space of sequences. Our score uses the density function of the push-forward measure of $x = G_\theta(z)$, which we call the sequence density. Consequentially, our score depends only on the decoder network, not the encoder, and can potentially be applied to other generative models like GANs or diffusion models.

**Why use the sequence density function?** We argue that for a well-trained decoder network, the density should be higher in areas of the sequence space close to the training data. To see why, consider a decoding model $G_\theta$ trained on a dataset $\{(z_i, x_i)\}_{i=1}^n$. The average loss (L) at $z$ is

$$\ell(G_\theta(z)) = \mathbb{E}_{x|E_\theta(x)=z} L(G_\theta(z), x). \quad (6)$$

As the training process is designed to minimize the population loss: $\mathbb{E}\ell(G_\theta(z))$, if successful, we hypothesize that the distribution of $G_\theta(z)$ puts higher weight in the areas where $\ell(G_\theta(z))$ is low. Since we expect most of the training data to be valid and to achieve low expected loss, the sequence density should put higher weight on the latent space valid set. In, Section 3.2 we provide an empirical validation for this hypothesis, for Examples 1.1 to 1.3. We highlight that this relationship between the valid set and the sequence density depends on how well the decoder fits the data.

### 3.1. Derivation of LES

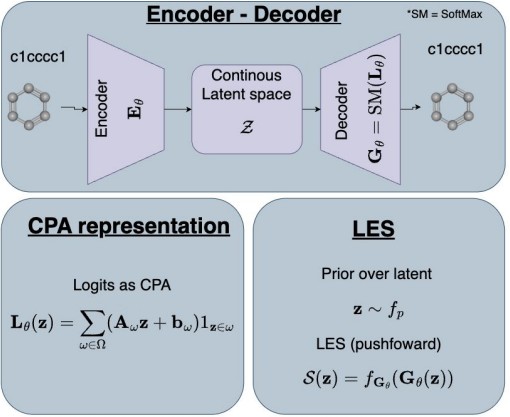

*Figure 2.* Derivation of LES. The decoder network ($G_\theta$), which maps from the latent space to the output space, is assumed to be the composition of a softmax operation over a continuous piecewise affine (CPA) spline operator. LES is the density of a random variable ($z$) in the latent space, under the decoder transformation. Calculating LES only requires a pre-trained decoder.

**Analytical formula for LES** DGNs for discrete sequences typically output a matrix of logits, transformed into normalized scores by the softmax function:

$$G_\theta(z) = \text{Softmax}(L_\theta(z)). \quad (7)$$

$L_\theta(z)$ is a $D \times L$ logits matrix (D - vocabulary size, L - sequence length) and Softmax is the softmax operation applied to every column of $L_\theta(z)$. In order to avoid a violation of the assumption that $G_\theta$ is bijective, we extend the function's output to include the normalizing constant for each column. We find that parametrizing the output to include the inverse of the normalizing constant (Equation (8)), helps in avoiding numerical instabilities that are caused by the constant being potentially very large. With this formulation, we can now derive the sequence density function.

**Theorem 3.1** (DGN sequence density)**.** *Let*

$$G_\theta(z) = \left( p_z^{(1)}, (c_z^{(1)})^{-1}, \ldots, p_z^{(L)}, (c_z^{(L)})^{-1} \right) = x_z \quad (8)$$

*where $p_z^{(i)} = Softmax(L_\theta(z))_{.i}$ and $c_z^{(i)} = \sum_{j=1}^D e^{L_\theta(z)_{ji}}$. Assume that $L_\theta$ is bijective and can be expressed as a CPA (Equation (5)), and that $z \sim p_z$, then the density function of $G_\theta(z)$ is given by:*

$$f_p(z) \sqrt{\det \left( \sum_{i=1}^L (A_i^\dagger)^T (B_i)^T B_i A_i^\dagger \right)} \quad (9)$$

*for*

$$B_i = \left( diag \left( \frac{1}{(p_z^{(i)})_1}, \ldots, \frac{1}{(p_z^{(i)})_D} \right), -1 \frac{1}{c_z^{(i)}} \right)^T \quad (10)$$

$$A_i^\dagger = \left( A_\omega^{(1)}, \ldots, A_\omega^{(L)} \right)^\dagger_{(i \cdot D):(i+1 \cdot D).}, \quad (11)$$

*where $\left( A_\omega^{(1)}, \ldots, A_\omega^{(L)} \right)^\dagger$ is the Moore–Penrose inverse of $\left( A_\omega^{(1)}, \ldots, A_\omega^{(L)} \right)$, and $f_p$ is the density function of $p_z$.*

The proof is provided in Appendix A. We define LES to be the logarithm of the determinant term,

$$S(z) = \log \left( \sqrt{\det \left( \sum_{i=1}^L (A_i^\dagger)^T (B_i)^T B_i A_i^\dagger \right)} \right), \quad (12)$$

as the contribution of the prior is negligible in magnitude in all the decoders we study.

*Remark* 3.2. LES can be calculated directly without assuming the decoder logits follow a CPA function (Ben-Israel, 1999). However, using the expessions derived in Equation (12) has two computational benefits for calculating the derivative of LES. First, using Jacobi's formula and observing that $\sum_{i=1}^L (A_i^\dagger)^T (B_i)^T B_i A_i^\dagger$ is a quadratic formula of the softmax probabilites, we can calculate the derivative of LES in closed form. Second, by the CPA assumption, the matrices $A_\omega^{(i)}$ are a constant function of $z$ and therefore $\frac{\partial A_\omega^{(i)}}{\partial z} = 0$. As a result, we avoid the need to calculate the hessian of the decoder when taking the derivative of LES.

**Limitations of Theorem 3.1** Our derivation relies on the decoder logits being (i) a CPA operator and (ii) bijective between the latent space and the generated manifold in the ambient space. We argue that (i) is not a restrictive assumption, as approximation theory has already demonstrated that any continuous model can be approximated by a CPA network. Therefore, one always recovers Theorem 3.1 even when using non-CPA models. Specifically, Daubechies et al. 2022 show that ReLU networks (which are CPA spline functions) can approximate refinable functions—a class of non-smooth functions—in addition to known results on other classes of functions (e.g., Weierstrass, polynomials, and analytic functions).

On the other hand, (ii) is a stronger assumption that practitioners should be mindful of, as it would invalidate LES as a meaningful metric for comparing different samples. For (ii) to be violated, i.e., for Equation (29) to be incorrect, the Lebesgue measure of the set $C_{\mathbf{G}_\theta} = \{z|\ \exists z^*; \mathbf{G}_\theta(z) = \mathbf{G}_\theta(z^*)\}$ must be larger than 0 (see Lemma A.2 for proof). This would suggest some degeneracy in the decoder function, where large regions of the latent space map to the same output, resulting in a zero gradient of the decoder with respect to its input. However, we believe this is rare in practice. In our experiments, where we compute the gradient of the decoder with respect to the input to calculate LES, we did not encounter instances where the gradient was zero.

Although we do not formally validate (ii) (the bijectivity assumption), we argue through our empirical analysis in Table 22, conducted across 22 VAEs (including pre-trained models), that (ii) may hold or, at the very least, serve as a reasonable approximation for real-world VAEs.

Lastly, it is crucial to highlight that the ability of LES to detect out-of-distribution data is closely tied to the decoder's capacity to accurately model the data. Although Theorem 3.1 holds for poorly trained decoders under the given conditions, we advise against relying on LES in such cases.

**Computing LES** LES is a function of the matrices $B_i$ and $A_\omega$. The matrices $B_i$ are a function of the logits and can be calculated using a single forward run of $\mathbf{G}_\theta$. The matrix $A_\omega$ is equal to the derivative of $\mathbf{L}_\theta$ at $z$, and can therefore can be obtained using automatic differentiation ((Paszke et al., 2017)). To avoid the (pseudo) inversion of the matrix $A_\omega$ we can exploit the inverse function theorem which states that $J\mathbf{G}_\theta^{-1} = (J\mathbf{G}_\theta)^{-1}$ and we can take $\mathcal{S}(z) = -0.5 \log \det((A_\omega \frac{\partial \mathbf{G}_\theta(\mathbf{L}_\theta)}{\partial \mathbf{L}_\theta})(A_\omega \frac{\partial \mathbf{G}_\theta(\mathbf{L}_\theta)}{\partial \mathbf{L}_\theta})^T)$, where $\frac{\partial \mathbf{G}_\theta(\mathbf{L}_\theta)}{\partial \mathbf{L}_\theta}$ (derivative of the softmax w.r.t the logits) admits a simple closed form solution. Ideally, LES can be computed by performing all of the above calculations in parallel using a single forward call to the $\mathbf{G}_\theta$ network. In addition, the computation of the determinant is done via SVD on a square matrix whose dimension is the latent dimension

of the decoder ($d$) with complexity of $\mathcal{O}(d^3)$.

In Table 1 we provide the wall clock times for calculating LES for a batch of 20 latent vectors across all architectures and datasets studies in this work. LES is compared with the Bayesian uncertainty score proposed by Notin et al. 2021 (with the default configuration: 10 sampled models and 40 sampled outcomes), which was previously used to regularize LSO. We also compare with a *Likelihood* score:

$$\ell(z) = \max_x p_{\mathbf{G}_\theta}(\boldsymbol{X} = \boldsymbol{x}|\boldsymbol{Z} = \boldsymbol{z}) \qquad (13)$$

where $p_{\mathbf{G}_\theta}(z)$ is the distribution defined by the decoder softmax probabilities, which reflects the likelihood of the most likely $x$ conditioned on the latent vector $z$.

For 9 out of the 10 models, particularly when the latent dimension is larger (e.g., SMILES and SELFIES), LES is computed faster than the Uncertainty score, achieving reductions of up to 85% in some cases. As the Likelihood score requires only a single forward pass of the decoder, it offers a more computationally efficient alternative, which comes with some performance trade-offs (Sections 3.2 and 4).

*Table 1.* Wall clock times in seconds (lower is better) for calculating LES, the Bayesian uncertainty and the Likelihood scores for a sample of 20 latent vectors on a single A100 GPU.

| Dataset | Arch. (dim.) | LES | Uncertainty | Likelihood |
|---|---|---|---|---|
| | GRU (25) | 0.730 | 0.823 | 0.025 |
| Expressions | LSTM (25) | 0.164 | 0.857 | 0.049 |
| | Transformer (25) | 0.526 | 0.481 | 0.029 |
| | GRU (56) | 0.663 | 3.157 | 0.103 |
| SMILES | LSTM (56) | 0.696 | 3.990 | 0.123 |
| | Transformer (56) | 0.581 | 0.726 | 0.185 |
| | GRU (75) | 0.498 | 1.925 | 0.064 |
| SELFIES | LSTM (75) | 0.525 | 2.442 | 0.071 |
| | Transformer (75) | 0.451 | 0.583 | 0.085 |
| | Transformer (256) | 7.422 | 42.882 | 0.410 |
| **Average** | – | 1.226 | 5.787 | 0.114 |

### 3.2. Validating the Relationship Between LES and Valid Generation

To assess LES's ability to identify valid regions (as defined in Examples 1.1 to 1.3), we sample data points in the latent space using the twenty-two VAEs studied in Section 4. Specifically, we sample 500 data points from three distributions: train, prior ($\mathcal{N}(\mathbf{0}, \boldsymbol{I})$), and out-of-distribution ($\mathcal{N}(\mathbf{0}, \boldsymbol{I} \cdot 5)$). We decode each data point and determine if the decoded sequence is valid.

Identifying if a point in the latent space decodes into a valid sequence can be viewed as a classification problem, in which the different scores (i.e., LES or the Bayesian uncertainty score) provide (unnormalized) probabilities for a

*Table 2.* AUROC values (higher is better) for identifying valid data points within the latent space, across datasets and decoder architectures. Data points are sampled from the training data, the VAE prior (standard Gaussian), and out-of-distribution data (Gaussian with std of 5). LES achieves the best performance in most cases (18 out of 22) and on average.

| Dataset | Arch. | $\beta$ | LES | Prior | Uncertainty | Likelihood |
|---|---|---|---|---|---|---|
| SMILES | GRU | 0.05 | 0.93 | 0.09 | 0.85 | **0.94** |
| | | 0.1 | **0.94** | 0.12 | 0.84 | **0.94** |
| | | 1.0 | 0.91 | 0.16 | 0.87 | **0.92** |
| | LSTM | 0.05 | **0.99** | 0.07 | **0.99** | 0.98 |
| | | 0.1 | 0.89 | 0.21 | 0.89 | **0.90** |
| | | 1.0 | **0.97** | 0.12 | 0.95 | 0.96 |
| | Transformer | 0.05 | **0.93** | 0.14 | 0.84 | 0.92 |
| | | 0.1 | **0.94** | 0.14 | 0.87 | 0.93 |
| | | 1.0 | **0.97** | 0.10 | 0.89 | 0.95 |
| Expressions | GRU | 0.05 | **0.96** | 0.38 | **0.96** | 0.88 |
| | | 0.1 | **0.94** | 0.42 | **0.94** | 0.80 |
| | | 1.0 | **0.94** | 0.57 | **0.94** | 0.89 |
| | LSTM | 0.05 | **0.96** | 0.38 | 0.90 | **0.96** |
| | | 0.1 | **0.95** | 0.37 | 0.91 | **0.95** |
| | | 1.0 | **0.95** | 0.56 | 0.91 | 0.91 |
| | Transformer | 0.05 | **0.91** | 0.43 | 0.86 | 0.90 |
| | | 0.1 | **0.91** | 0.53 | 0.87 | 0.89 |
| | | 1.0 | 0.86 | 0.70 | **0.92** | **0.92** |
| SELFIES | Transformer | 0.05 | **1.0** | 0.02 | 0.99 | 0.97 |
| | | 0.1 | **0.99** | 0.03 | 0.96 | 0.98 |
| | | 1.0 | **0.95** | 0.06 | 0.85 | 0.93 |
| SELFIES ((Maus et al., 2022)) | Transformer | – | **0.75** | 0.69 | 0.33 | 0.70 |
| | Average | | **0.93** | 0.29 | 0.88 | 0.91 |

sequence being valid. We measure the performance of these scores using the AUROC metric. Besides LES, the Bayesian uncertainty score and the Likelihood score, we add three additional baseline scores for comparison. The first is the density of a standard Gaussian (**Prior**), which is the distribution the latent vectors are regularized to follow during VAE training. The second is the polarity score (**Polarity**) (Humayun et al., 2022), based only on the derivative of the decoder logits with respect to the latent vector, which shows the gains due to accounting for the softmax non-linearity in the derivation of LES (Theorem 3.1). We also consider the average distance to the closest three data points within a random sample of 1000 points from the training data in the latent space (**Train distances**).

The results are shown in Tables 2 and 22. LES provides the best performance in 18 out of the 22 VAEs in this analysis, and in all cases provides a clear signal for identifying valid regions, as indicated by AUROC values that are at least 0.75. This is while being much faster to compute than the Uncertainty score (Table 1) and without the need to store a potentially large array of latent vectors.

## 4. LES-Constrained LSO

In Section 3.2 we showed that LES is a robust score that obtains higher values in the latent space valid set (Equation (4)). Furthermore, LES is differentiable, which means

it can easily be used to constrain any optimization problem. Therefore, we propose adding an explicit constraint to Equation (3), encouraging the solution to achieve a high LES value. We modify Algorithm 1 by penalizing step (2):

$$z^{\text{new}} = \arg\max_{z} \mathcal{A}(\hat{f}(z)) + \lambda \mathcal{S}(z). \qquad (14)$$

### 4.1. Experimental Setup

**VAE models** To evaluate the effectiveness of LES as a regularization method for LSO, we trained twenty-two VAEs, focusing on varying the decoder architectures and the $\beta$ parameter, which controls the trade-off between reconstruction loss and alignment with the prior (KL divergence term). All models use a convolutional encoder based on the architecture proposed by Kusner et al. (2017), and were trained for 300 epochs using the Adam optimizer (Kingma, 2014) with a learning rate of 1e-3 and batch size of 256.

The VAEs for the Expressions dataset, sourced from Kusner et al. (2017), were trained on 80k data points with a latent dimension of 25. The SMILES VAEs were trained on the ZINC250k dataset, consisting of approximately 250k drug-like molecules in SMILES format. Following Kusner et al. (2017) and Notin et al. (2021), a latent dimension of 56 was used. For the SELFIES VAEs, we used a subset of approximately 200k molecules from the ZINC250k dataset that passed a set of quality filters (Walters, 2019), using the SELFIES representation (Krenn et al., 2020), with a latent dimension of 75. Additionally, the pre-trained VAE by Maus et al. (2022) had a latent dimension of 256.

**LSO setup** We begin by training a single-task Gaussian Process on an initial dataset. Each sample is mapped to a latent space and paired with its true objective value. For Expressions and SMILES VAEs, we use 500 data points. For SELFIES VAEs, we use 1500. Across all tasks, we generate 500 candidate solutions per problem, aligned with prior work (Kusner et al., 2017; Notin et al., 2021) and reflective of real-world wet-lab constraints (Gao et al., 2022). Solutions are proposed in batches of 20.

For the SELFIES VAEs (both from Maus et al. (2022) and those trained by us), we employ a deep kernel that reduces the latent space to 12 dimensions before fitting the Gaussian Process. To mitigate vanishing gradients, we use log expected improvement (Ament et al., 2024) as our acquisition function, which is sequentially maximized.

**Optimization tasks** The Expressions dataset consists of arithmetic expressions that are functions of a single variable (e.g., $\sin(x)$, $1 + x * x$). Our objective is to find an expression that approximates `1/3 + x + sin(x * x)`, as described by Kusner et al. (2017). The optimization target

is defined as $\mathcal{M}(x) = -\log(1+\text{MSE}(x))$, where $\text{MSE}(x)$ is the mean-squared error between the expression $x$ and `1/3 + x + sin(x * x)`, evaluated over the range -10 to 10 using a grid of 1000 equally spaced points.

For the SMILES dataset, our goal is to maximize the octanol-water partition coefficient, which is calculated using the prediction model developed by Wildman & Crippen (1999). In the case of the SELFIES dataset, following Maus et al. (2022), we focus on three objectives: Perindopril MPO, Ranolazine MPO, and Zaleplon MPO, all of which are part of the Guacamol benchmarks (Brown et al., 2019). While the SELFIES syntax is 100% robust, we consider only solutions that pass a set of quality filters for evaluation (see Example 1.3 for more details).

**LES-constrained LSO**    For ease of implementation and numerical stability, when applying LES regularization, we adopt a simple optimization procedure in which the acquisition function is optimized using 10 steps of normalized gradient ascent. This is because we empirically find that the norm of the derivative of the constraint (i.e., $\mathcal{S}(z)$) is typically much larger than the norm of the derivative of the acquisition function. As a result, using the gradient ascent update rule $z^{(i+1)} = \partial\mathcal{A}(\hat{f}(z^{(i)})) + \lambda\partial\mathcal{S}(z^{(i)})$ leads to a numerically unstable optimization process. To address this issue, we propose the following update rule:

$$z^{(i+1)} = \frac{\partial\mathcal{A}(\hat{f}(z^{(i)}))}{\|\partial\mathcal{A}(\hat{f}(z^{(i)}))\|_2} + \lambda\frac{\partial\mathcal{S}_\rho(z^{(i)})}{\|\partial\mathcal{S}_\rho(z^{(i)})\|_2}. \quad (15)$$

We set $\lambda = 0.05$ for Expressions, $\lambda = 0.1$ for our SELFIES models, and $\lambda = 0.5$ for SMILES and the pre-trained SELFIES-VAE. For Ranolazine MPO with pre-trained SELFIES-VAE, we use $\lambda = 0.1$ to prevent over-regularization, as regularized methods already yield a high percentage of valid solutions (Table 24). We select these values because, without regularization, the SMILES and pre-trained SELFIES models tend to produce a lower percentage of valid solutions (Table 24). Based on our findings, we suggest $\lambda = 0.5$ as a reasonable default value, while leaving the exploration of an optimal choice for future work.

**Benchmark methods**    We compare our LES-constrained method (**LES**) with five alternative approaches for optimizing the acquisition function. First, we evaluate a non-regularized version of Equation (15), where $\lambda = 0$ (**LSO (GA)**), and a two regularization methods that uses the prior density (i.e., $\ell_2$ regularization) and likelihood (Equation (13)) instead of LES (**Prior**, and **Likelihood** respectively), using similar $\lambda$ values described above.

Additionally, we consider optimizing the acquisition function using the Limited-memory BFGS method within a symmetric hypercube centered at 0 (**LSO (L-BFGS)**), which is

the default approach in the `BoTorch` package (Balandat et al., 2020). Since recent state-of-the-art LSO pipelines (Maus et al., 2022; Lee et al., 2024) have utilized trust regions centered around the best observed value (Eriksson et al., 2019), we also compare with this approach (**TuRBO**). Lastly, we implement the Uncertainty Censoring (**UC**) method proposed by Notin et al. (2021), which suggests early stopping of the optimization when the estimated uncertainty exceeds a certain threshold. For this threshold, we use the 99th percentile of the observed uncertainty values in the training data, as recommended by Notin et al. (2021).

**Hyperparameters**    We calibrate the step size, which affects **LES**, **UC**, **Prior**, **Likelihood**, and **LSO (GA)**, to ensure our gradient ascent procedure (with $\lambda = 0$) improves the acquisition function values across different initializations. The same step size is applied to all models within the same dataset: Expressions = 0.8, SMILES = 0.003, SELFIES = 0.03, and SELFIES pre-trained = 0.3. The LSO (L-BFGS) method has a single hyperparameter, the facet length, which is set to 5. For TuRBO, there are three primary hyperparameters: the initial length, which we set to 0.8, along with the success and failure tolerances, determining when to expand or shrink the trust region, set at 10 and 2, respectively. An ablation study is provided in Appendix B.

### 4.2. Results

**Optimization results**    The experimental results, presenting the average across 10 independent LSO runs for the top 20 and best solutions found, are summarized in Tables 20 and 21 respectively. In both cases, LES achieves the average best performance most frequently (22 and 17 out of 30 times, respectively). Furthermore, LES outperforms other methods in both the average ranking across LSO tasks and the frequency with which it falls within one standard deviation of the best-performing method (Table 3). These findings demonstrate that using LES as a regularization technique generally enhances optimization performance.

Figures 3 and 4 show the cumulative average top-20 and the best objective values during BO with the pre-trained SELFIES-VAE (Maus et al., 2022), respectiveley. Results align with (Maus et al., 2022) under our realistic evaluation budget. For the average of the top-20 solutions on the Ranolazine MPO and Zaleplon MPO tasks, LES outperforms TuRBO on valid solutions but underperforms overall, highlighting that LES effectively constrains the optimzation.

Table 24 shows the percentage of valid solutions found by each method across datasets and VAEs. LES improves upon the non-regularized version of gradient ascent by 7% on average and upon TuRBO and LSO (L-BFGS) by 24% and 36% on average respectively. While UC achieves a 2% higher percentage of valid solutions on average, we show

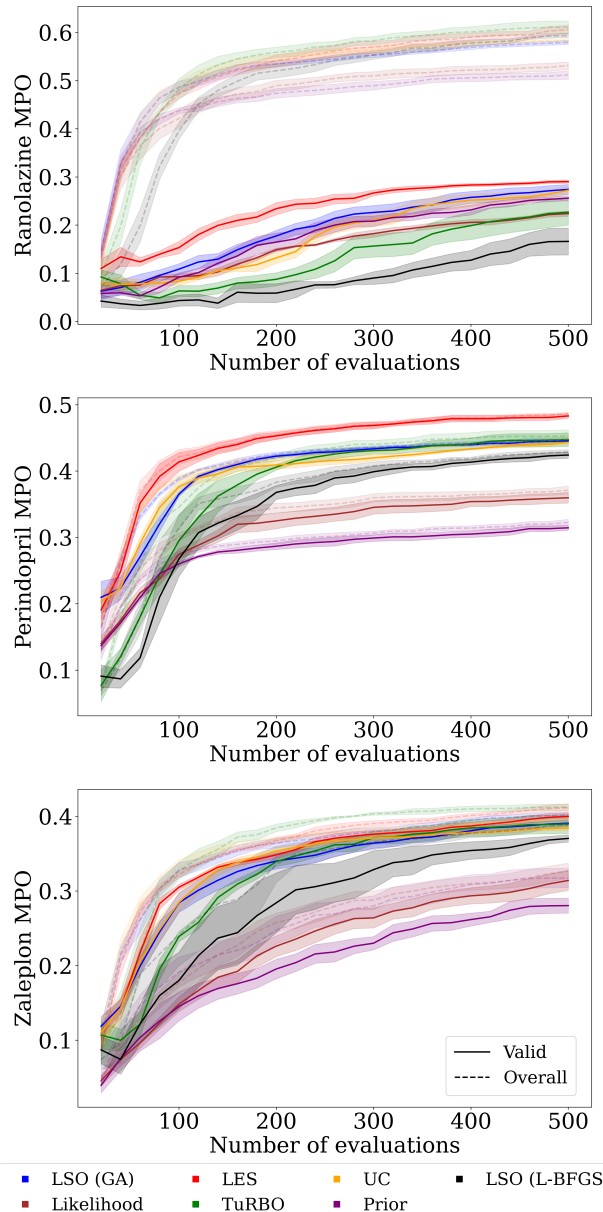

*Figure 3.* Cumulative objective for the top-20 solutions found during Bayesian optimization with the pre-trained SELFIES-VAE (27). Each method is shown in a distinct color. Solid lines represent solutions passing quality filters, while dashed lines include all evaluations. LES outperforms all baselines on Ranolazine MPO and Perindopril MPO, achieving competitive results on Zaleplon MPO.

*Table 3.* Summary metrics for 30 Bayesian Optimization experiments from Tables 20 and 21. We report the average rank (lower is better) and the count of times each method is within one standard deviation of the best. Results cover both the best solution and the top-20 average. LES outperforms alternatives on both metrics.

| Method | Top 1 | | Top 20 | |
|---|---|---|---|---|
| | Avg Rank | # within std | Avg Rank | # within std |
| LES | **2.15** | **19** | **1.9** | **22** |
| Likelihood | 2.76 | 16 | 2.53 | 13 |
| LSO (GA) | 3.15 | 8 | 2.87 | 7 |
| Prior | 3.86 | 10 | 3.8 | 5 |
| UC | 4.17 | 5 | 4.3 | 3 |
| TuRBO | 5.7 | 2 | 6.1 | 0 |
| LSO (L-BFGS) | 6.17 | 1 | 6.4 | 0 |

in Table 23 that for the Expressions datasets, where UC excels, the number of valid produced by LES solutions can be increased by setting a higher $\lambda$ value. However, this did not improve optimization performance.

## 5. Discussion

We proposed LES to mitigate over-exploration in latent space optimization (LSO). LES is differentiable and fully parallelizable. Extensive evaluations demonstrate that incorporating LES as a penalty in LSO consistently enhances solution quality and objective outcomes. Moreover, LES outperforms alternative regularization techniques, proving to be the most robust across diverse datasets and varying definitions of validity. In addition, LES has only a single hyperparameter (the regularization strength), and we observe empirically that deploying LES can provide significant performance gains. We therefore believe LES offers a powerful approach for discovering more realistic solutions, when the criteria for realism are difficult to define or validate.

While LES is fully parallelizable, it requires the calculation of the derivative of the decoder as well as the determinant of the change-of-variables term, which can be computationally expensive. This step can become a bottleneck when the size of the output and the latent dimension are both large. It is left for future work to develop a fast approximation for this operation in order to enable the use of LES in applications involving large generative models.

## Acknowledgements

Ronen and Yu gratefully acknowledge partial support from NSF grant DMS-2413265, NSF grant DMS 2209975, NSF grant 2023505 on Collaborative Research: Foundations of Data Science Institute (FODSI), the NSF and the Simons Foundation for the Collaboration on the Theoretical Foundations of Deep Learning through awards DMS-2031883 and

814639, NSF grant MC2378 to the Institute for Artificial CyberThreat Intelligence and OperatioN (ACTION), and NIH (DMS/NIGMS) grant R01GM152718.

Humayun and Baraniuk gratefully acknowledge the support from NSF grants CCF1911094, IIS-1838177, and IIS-1730574; ONR grants N00014-18-12571, N00014-20-1-2534, and MURI N00014-20-1-2787; AFOSR grant FA9550-22-1-0060; and a Vannevar Bush Faculty Fellowship, ONR grant N00014-18-1-2047.

## Impact Statement

This work seeks to improve the practicality of latent space optimization, a method first used in drug discovery. It brings no new societal implications beyond those already associated with LSO.

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

## A. Proofs

**Lemma A.1.** *Let $f_\theta$ be a DGN as defined in Equation (8) and assume that $f_\theta$ can be expressed as a CPA (Equation (5)) and is inevitable, then*

$$Jf_\theta^{-1}(\boldsymbol{x}) = \left( \begin{bmatrix} \boldsymbol{B}_1 & \cdots & \boldsymbol{0} \\ \vdots & \ddots & \vdots \\ \boldsymbol{0} & \cdots & \boldsymbol{B}_L \end{bmatrix} \boldsymbol{A}_\omega^\dagger \right)^T, \tag{16}$$

*where $\boldsymbol{A}_\omega^\dagger$ is the Moore–Penrose inverse of the slope matrix, at the knot whose image constrains $\boldsymbol{x}$, and*

$$\boldsymbol{B}_i = \left( diag \left( \frac{1}{(\boldsymbol{p}_{\boldsymbol{z}}^{(i)})_1}, \ldots, \frac{1}{(\boldsymbol{p}_{\boldsymbol{z}}^{(i)})_D} \right), -\boldsymbol{1}\frac{1}{c_{\boldsymbol{z}}^{(i)}} \right)^T. \tag{17}$$

*Proof.* First we write

$$f_\theta(\boldsymbol{z}) = \mathrm{Softmax}_+(\ell_\theta(\boldsymbol{z})), \tag{18}$$

Where $\mathrm{Softmax}_+$ is the extension of the column wise Softmax function to include the normalizing constants. Specifically, for L by D $\ell_\theta(\boldsymbol{z})$ matrix, we have

$$\mathrm{Softmax}_+(\ell_\theta(\boldsymbol{z})) = \left( \boldsymbol{p}_{\boldsymbol{z}}^{(1)}, (c_{\boldsymbol{z}}^{(1)})^{-1}, \ldots, \boldsymbol{p}_{\boldsymbol{z}}^{(L)}, (c_{\boldsymbol{z}}^{(L)})^{-1} \right) = \boldsymbol{x}_{\boldsymbol{z}}, \tag{19}$$

with $\boldsymbol{p}_{\boldsymbol{z}}^{(i)} = (\frac{e^{\ell_\theta(\boldsymbol{z})_{1i}}}{c_{\boldsymbol{z}}^{(i)}})$, and $c_{\boldsymbol{z}}^{(i)} = \sum_{j=1}^D \exp(\ell_\theta(\boldsymbol{z})_{ji})$.
Next,

$$f_\theta^{(-1)}(\boldsymbol{x}) = \ell_\theta^{-1}(\mathrm{Softmax}_+^{-1}(\boldsymbol{x})) \tag{20}$$

A direct calculation yields,

$$\mathrm{Softmax}_+^{-1}(\boldsymbol{x}) = \left( \log(\boldsymbol{p}_{\boldsymbol{z}}^{(1)}) + \log(c_{\boldsymbol{z}}^{(1)}), \ldots, \log(\boldsymbol{p}_{\boldsymbol{z}}^{(L)}) + \log(c_{\boldsymbol{z}}^{(L)}) \right). \tag{21}$$

As we assume $\ell_\theta$ is bijective and can be written as

$$\ell_\theta(\boldsymbol{z}) = \sum_{\omega \in \Omega} (\boldsymbol{A}_\omega \boldsymbol{z} + \boldsymbol{b}_\omega) \, 1_{\boldsymbol{z} \in \omega}, \tag{22}$$

we have that

$$\ell_\theta^{-1}(\mathrm{Softmax}_+^{-1}(\boldsymbol{x})) = (\mathrm{Softmax}_+^{-1}(\boldsymbol{x}) - \boldsymbol{b}_\omega) \boldsymbol{A}_\omega^\dagger. \tag{23}$$

Lastly, as

$$\frac{\partial \mathrm{Softmax}_+^{-1}(\boldsymbol{x})}{\partial \boldsymbol{x}} = \begin{bmatrix} \boldsymbol{B}_1 & \cdots & \boldsymbol{0} \\ \vdots & \ddots & \vdots \\ \boldsymbol{0} & \cdots & \boldsymbol{B}_L \end{bmatrix}, \tag{24}$$

for

$$\boldsymbol{B}_i = \left( diag \left( \frac{1}{(\boldsymbol{p}_{\boldsymbol{z}}^{(i)})_1}, \ldots, \frac{1}{(\boldsymbol{p}_{\boldsymbol{z}}^{(i)})_D} \right), -\boldsymbol{1}\frac{1}{c_{\boldsymbol{z}}^{(i)}} \right)^T. \tag{25}$$

we obtain the final result. $\qquad\square$

*Proof of Theorem 3.1.* First, we note that by our invertibility assumption we have that $\mathbb{P}(\boldsymbol{x} \in W) = \mathbb{P}(\boldsymbol{z} \in f_\theta^{(-1)}(W))$. We then proceed with a direct calculation

$$\mathbb{P}(\boldsymbol{x} \in W) = \mathbb{P}(\boldsymbol{z} \in f_\theta^{(-1)}(W)) \tag{26}$$

$$= \sum_{\omega \in \Omega} \mathbb{P}(\boldsymbol{z} \in (f_\theta^{(-1)}(W) \cap \omega)) \tag{27}$$

$$= \sum_{\omega \in \Omega} \int_{f_\theta^{(-1)}(W) \cap \omega} f_{\boldsymbol{z}}(\boldsymbol{z}) d\boldsymbol{z} \tag{28}$$

$$= \sum_{\omega \in \Omega} \int_{W \cap f_\theta(\omega)} f_{\boldsymbol{z}}(f_\theta^{(-1)}(\boldsymbol{x})) \sqrt{\det \left( J f_\theta^{(-1)}(\boldsymbol{x}) J f_\theta^{(-1)}(\boldsymbol{x})^T \right)} d\boldsymbol{x} \tag{29}$$

$$= \int_W \sum_{\omega \in \Omega} f_{\boldsymbol{z}}(f_\theta^{(-1)}(\boldsymbol{x})) \sqrt{\det \left( J f_\theta^{(-1)}(\boldsymbol{x}) J f_\theta^{(-1)}(\boldsymbol{x})^T \right)} 1_{\{\boldsymbol{x} \in f_\theta(\omega)\}} d\boldsymbol{x}. \tag{30}$$

Using Lemma A.1, we get that the volume element is

$$J f_\theta^{(-1)}(\boldsymbol{x}) J f_\theta^{(-1)}(\boldsymbol{x})^T = \left( \begin{bmatrix} \boldsymbol{B}_1 & \cdots & \boldsymbol{0} \\ \vdots & \ddots & \vdots \\ \boldsymbol{0} & \cdots & \boldsymbol{B}_{\mathrm{L}} \end{bmatrix} \boldsymbol{A}_\omega^\dagger \right)^T \left( \begin{bmatrix} \boldsymbol{B}_1 & \cdots & \boldsymbol{0} \\ \vdots & \ddots & \vdots \\ \boldsymbol{0} & \cdots & \boldsymbol{B}_{\mathrm{L}} \end{bmatrix} \boldsymbol{A}_\omega^\dagger \right) \tag{31}$$

$$\left( (\boldsymbol{A}_\omega^\dagger)^T \begin{bmatrix} \boldsymbol{B}_1^T & \cdots & \boldsymbol{0} \\ \vdots & \ddots & \vdots \\ \boldsymbol{0} & \cdots & \boldsymbol{B}_{\mathrm{L}}^T, \end{bmatrix} \right) \left( \begin{bmatrix} \boldsymbol{B}_1 & \cdots & \boldsymbol{0} \\ \vdots & \ddots & \vdots \\ \boldsymbol{0} & \cdots & \boldsymbol{B}_{\mathrm{L}} \end{bmatrix} \boldsymbol{A}_\omega^\dagger \right) \tag{32}$$

$$= \sum_{i=1}^{\mathrm{L}} (\boldsymbol{A}_i^\dagger)^T (\boldsymbol{B}_i)^T \boldsymbol{B}_i \boldsymbol{A}_i^\dagger, \tag{33}$$

where $\boldsymbol{A}_i^\dagger = \left( \boldsymbol{A}_\omega^{(1)}, \ldots, \boldsymbol{A}_\omega^{(L)} \right)_{(i \cdot \mathrm{D}):(i+1 \cdot \mathrm{D}).}^\dagger$. $\qquad\square$

**Lemma A.2.** *Let $f : \mathcal{Z} \to \mathcal{X}$ be a function and define $f^\dagger : \mathcal{X} \to \mathcal{Z}$ as $f^\dagger(x) \in \{z : f(z) = x\}$. Let $\mu$ be the Lebesgue measure and assume that $\mu(\{z; \exists z' \text{ s.t. } f(z) = f(z')\}) = 0$, then for every $B \subseteq \mathcal{Z}$ we have*

$$\mu(\{z; f(z) \in B\}) = \mu(f^\dagger(B)) \tag{34}$$

*Proof.* We proceed with direct calculation

$$\mu(\{z; f(z) \in B\}) \le \mu(f^\dagger(B)) + \mu(\{z; \exists z' \text{ s.t. } f(z) = f(z')\}) \tag{35}$$

Now assume that $\mu(\{z; \exists z' \text{ s.t. } f(z) = f(z')\}) = 0$, we have that $\mu(\{z; f(z) \in B\}) \le \mu(f^\dagger(B))$. The other direction follows immediately from the definition of $f^\dagger$. $\qquad\square$

Lemma A.2 implies that Equation (29) can still hold under the assumption that $\mu(\{z; \exists z' \text{ s.t. } f_\theta(z) = f_\theta(z')\}) = 0$.

# B. Ablation studies

### B.1. TuRBO hyper-parameters

We provide an ablation study of the initial length and success and failure tolerances. For each setup, the values reported in the main paper are highlighted in bold.

*Table 4.* Ablation study for the initial length, success tolerance and failure tolerance for the TuRBO method. Average across 10 independent runs, of the top 20 best values across datasets, architectures. Column names indicate the length/success/fail values. Results from the main paper are in bold.

| | Architecture | $\beta$ | 0.8/10/10 | 0.8/10/2 | 0.8/2/10 | 0.8/2/2 | 1.6/10/10 | 1.6/10/2 | 1.6/2/10 | 1.6/2/2 |
|---|---|---|---|---|---|---|---|---|---|---|
| **Expressions** | GRU | 0.05 | -2.03 (0.12) | **-2.1 (0.11)** | -2.0 (0.09) | -2.0 (0.11) | -1.93 (0.09) | -1.92 (0.08) | -1.91 (0.06) | -1.93 (0.07) |
| | | 0.1 | -2.18 (0.15) | **-2.14 (0.18)** | -2.11 (0.2) | -2.14 (0.2) | -1.98 (0.13) | -2.05 (0.12) | -2.0 (0.17) | -2.09 (0.22) |
| | | 1 | -1.39 (0.12) | **-1.34 (0.05)** | -1.56 (0.11) | -1.61 (0.11) | -1.36 (0.08) | -1.42 (0.11) | -1.45 (0.12) | -1.38 (0.12) |
| | LSTM | 0.05 | -1.45 (0.09) | **-1.59 (0.13)** | -1.63 (0.12) | -1.49 (0.1) | -1.46 (0.08) | -1.52 (0.08) | -1.51 (0.12) | -1.31 (0.07) |
| | | 0.1 | -1.31 (0.12) | **-1.32 (0.13)** | -1.37 (0.11) | -1.31 (0.11) | -1.31 (0.1) | -1.36 (0.13) | -1.36 (0.11) | -1.35 (0.09) |
| | | 1 | -2.16 (0.06) | **-2.22 (0.08)** | -2.23 (0.08) | -2.23 (0.08) | -2.13 (0.07) | -2.16 (0.06) | -2.15 (0.05) | -2.08 (0.06) |
| | Transformer | 0.05 | -2.07 (0.12) | **-2.16 (0.09)** | -2.03 (0.17) | -2.15 (0.14) | -1.96 (0.13) | -1.74 (0.13) | -2.09 (0.1) | -1.79 (0.11) |
| | | 0.1 | -1.45 (0.13) | **-1.39 (0.12)** | -1.56 (0.13) | -1.38 (0.13) | -1.32 (0.11) | -1.43 (0.17) | -1.4 (0.1) | -1.33 (0.12) |
| | | 1 | -1.96 (0.11) | **-1.77 (0.12)** | -1.81 (0.11) | -1.85 (0.09) | -1.85 (0.11) | -1.76 (0.08) | -1.78 (0.08) | -1.75 (0.09) |
| **SELFIES** | Transformer (pdop) | 0.05 | 0.02 (0.01) | **0.04 (0.01)** | 0.04 (0.01) | 0.04 (0.01) | 0.09 (0.01) | 0.09 (0.03) | 0.08 (0.01) | 0.09 (0.02) |
| | | 0.1 | 0.01 (0.0) | **0.01 (0.0)** | 0.02 (0.01) | 0.02 (0.01) | 0.03 (0.01) | 0.08 (0.03) | 0.04 (0.02) | 0.05 (0.02) |
| | | 1 | 0.24 (0.01) | **0.23 (0.02)** | 0.21 (0.02) | 0.24 (0.01) | 0.27 (0.01) | 0.26 (0.01) | 0.28 (0.01) | 0.26 (0.01) |
| | Transformer (rano) | 0.05 | 0.06 (0.01) | **0.06 (0.0)** | 0.06 (0.01) | 0.06 (0.01) | 0.04 (0.01) | 0.02 (0.0) | 0.04 (0.0) | 0.03 (0.01) |
| | | 0.1 | 0.03 (0.01) | **0.03 (0.01)** | 0.03 (0.0) | 0.02 (0.0) | – | – | – | – |
| | | 1 | – | **–** | 0.08 (0.0) | – | 0.05 (0.01) | 0.07 (0.01) | 0.07 (0.0) | – |
| | Transformer (zale) | 0.05 | 0.02 (0.01) | **0.04 (0.01)** | 0.02 (0.01) | 0.04 (0.01) | 0.07 (0.01) | 0.07 (0.02) | 0.06 (0.01) | 0.06 (0.01) |
| | | 0.1 | 0.04 (0.01) | **0.04 (0.01)** | 0.04 (0.01) | 0.03 (0.01) | 0.01 (0.0) | 0.02 (0.0) | 0.09 (0.0) | 0.04 (0.0) |
| | | 1 | 0.16 (0.02) | **0.17 (0.02)** | 0.16 (0.02) | 0.17 (0.02) | 0.18 (0.02) | 0.16 (0.02) | 0.18 (0.02) | 0.19 (0.02) |
| **SELFIES (27)** | Transformer (pdop) | 1 | 0.44 (0.01) | **0.44 (0.01)** | 0.44 (0.01) | 0.44 (0.01) | 0.45 (0.0) | 0.45 (0.01) | 0.44 (0.01) | 0.44 (0.01) |
| | Transformer (rano) | 1 | 0.20 (0.02) | **0.22 (0.02)** | 0.21 (0.01) | 0.20 (0.01) | 0.23 (0.01) | 0.22 (0.01) | 0.22 (0.01) | 0.23 (0.01) |
| | Transformer (zale) | 1 | 0.37 (0.01) | **0.37 (0.01)** | 0.36 (0.01) | 0.37 (0.01) | 0.39 (0.01) | 0.38 (0.01) | 0.39 (0.01) | 0.39 (0.01) |
| **SMILES** | GRU | 0.05 | 0.89 (0.19) | **0.82 (0.15)** | 0.64 (0.2) | 0.89 (0.14) | 1.0 (0.14) | 0.98 (0.19) | 0.7 (0.24) | 1.07 (0.19) |
| | | 0.1 | 0.4 (0.22) | **0.41 (0.19)** | 0.63 (0.26) | 0.55 (0.14) | 0.83 (0.13) | 0.96 (0.33) | 0.69 (0.18) | 0.75 (0.18) |
| | | 1 | 1.44 (0.0) | **-0.16 (0.0)** | – | – | 0.14 (0.65) | – | – | 1.14 (0.0) |
| | LSTM | 0.05 | 1.18 (0.25) | **1.54 (0.41)** | 1.1 (0.15) | 1.1 (0.36) | 0.86 (0.27) | 1.05 (0.21) | 1.07 (0.29) | 1.22 (0.27) |
| | | 0.1 | – | **–** | – | – | 0.21 (0.0) | – | – | – |
| | | 1 | 0.61 (0.16) | **0.52 (0.26)** | 0.48 (0.13) | 0.7 (0.23) | 0.67 (0.23) | 1.13 (0.16) | 0.97 (0.28) | 0.88 (0.23) |
| | Transformer | 0.05 | 0.88 (0.18) | **0.97 (0.19)** | 0.92 (0.14) | 0.67 (0.15) | 0.93 (0.17) | 1.16 (0.1) | 1.09 (0.11) | 0.97 (0.24) |
| | | 0.1 | 0.42 (0.12) | **0.62 (0.14)** | 0.44 (0.16) | 0.73 (0.14) | 0.91 (0.1) | 0.6 (0.14) | 0.81 (0.14) | 0.7 (0.14) |
| | | 1 | 0.72 (0.18) | **0.48 (0.18)** | 0.61 (0.15) | 0.47 (0.16) | 0.99 (0.12) | 0.74 (0.15) | 0.83 (0.16) | 0.68 (0.15) |

*Table 5.* Ablation study for the initial length, success tolerance and failure tolerance for the TuRBO method. Average across 10 independent runs, of the best value across datasets, architectures. Column names indicate the length/success/fail values. Results from the main paper are in bold.

| | Architecture | $\beta$ | 0.8/10/10 | 0.8/10/2 | 0.8/2/10 | 0.8/2/2 | 1.6/10/10 | 1.6/10/2 | 1.6/2/10 | 1.6/2/2 |
|---|---|---|---|---|---|---|---|---|---|---|
| **Expressions** | GRU | 0.05 | -0.65 (0.11) | **-0.73 (0.12)** | -0.71 (0.09) | -0.61 (0.1) | -0.65 (0.12) | -0.59 (0.06) | -0.62 (0.11) | -0.72 (0.13) |
| | | 0.1 | -0.56 (0.05) | **-0.61 (0.07)** | -0.59 (0.07) | -0.71 (0.1) | -0.54 (0.04) | -0.58 (0.05) | -0.62 (0.04) | -0.63 (0.04) |
| | | 1 | -0.56 (0.05) | **-0.54 (0.05)** | -0.54 (0.06) | -0.6 (0.08) | -0.56 (0.06) | -0.61 (0.05) | -0.6 (0.04) | -0.59 (0.05) |
| | LSTM | 0.05 | -0.46 (0.03) | **-0.43 (0.02)** | -0.43 (0.02) | -0.38 (0.02) | -0.43 (0.02) | -0.43 (0.02) | -0.47 (0.04) | -0.4 (0.01) |
| | | 0.1 | -0.38 (0.04) | **-0.39 (0.0)** | -0.41 (0.01) | -0.42 (0.02) | -0.42 (0.02) | -0.42 (0.02) | -0.42 (0.02) | -0.42 (0.02) |
| | | 1 | -0.96 (0.09) | **-0.86 (0.0)** | -0.86 (0.0) | -1.01 (0.11) | -0.88 (0.01) | -0.98 (0.06) | -0.92 (0.04) | -0.88 (0.01) |
| | Transformer | 0.05 | -0.39 (0.04) | **-0.44 (0.02)** | -0.39 (0.04) | -0.4 (0.05) | -0.44 (0.02) | -0.39 (0.04) | -0.38 (0.04) | -0.42 (0.02) |
| | | 0.1 | -0.38 (0.04) | **-0.41 (0.02)** | -0.42 (0.02) | -0.42 (0.02) | -0.39 (0.04) | -0.41 (0.01) | -0.41 (0.02) | -0.37 (0.04) |
| | | 1 | -0.67 (0.1) | **-0.58 (0.1)** | -0.52 (0.08) | -0.62 (0.06) | -0.66 (0.11) | -0.62 (0.07) | -0.56 (0.05) | -0.54 (0.05) |
| **SELFIES** | Transformer (pdop) | 0.05 | 0.13 (0.02) | **0.15 (0.03)** | 0.17 (0.03) | 0.18 (0.03) | 0.23 (0.03) | 0.17 (0.04) | 0.22 (0.04) | 0.2 (0.04) |
| | | 0.1 | 0.08 (0.02) | **0.09 (0.03)** | 0.1 (0.03) | 0.09 (0.03) | 0.1 (0.03) | 0.12 (0.04) | 0.1 (0.03) | 0.14 (0.04) |
| | | 1 | 0.36 (0.02) | **0.31 (0.03)** | 0.36 (0.02) | 0.34 (0.02) | 0.38 (0.01) | 0.38 (0.0) | 0.4 (0.01) | 0.36 (0.02) |
| | Transformer (rano) | 0.05 | 0.17 (0.02) | **0.2 (0.02)** | 0.2 (0.02) | 0.18 (0.02) | 0.11 (0.02) | 0.08 (0.01) | 0.12 (0.02) | 0.13 (0.02) |
| | | 0.1 | 0.09 (0.02) | **0.1 (0.01)** | 0.11 (0.02) | 0.1 (0.02) | 0.06 (0.01) | 0.05 (0.01) | 0.04 (0.01) | 0.06 (0.01) |
| | | 1 | 0.07 (0.02) | **0.05 (0.02)** | 0.11 (0.03) | 0.07 (0.02) | 0.16 (0.02) | 0.16 (0.03) | 0.11 (0.02) | 0.12 (0.02) |
| | Transformer (zale) | 0.05 | 0.11 (0.02) | **0.15 (0.03)** | 0.12 (0.02) | 0.16 (0.03) | 0.22 (0.03) | 0.23 (0.03) | 0.21 (0.04) | 0.19 (0.03) |
| | | 0.1 | 0.14 (0.01) | **0.16 (0.01)** | 0.15 (0.02) | 0.14 (0.01) | 0.1 (0.02) | 0.12 (0.02) | 0.13 (0.02) | 0.14 (0.03) |
| | | 1 | 0.36 (0.02) | **0.31 (0.03)** | 0.38 (0.02) | 0.36 (0.02) | 0.38 (0.01) | 0.34 (0.02) | 0.31 (0.03) | 0.38 (0.02) |
| **SELFIES (27)** | Transformer (pdop) | 1 | 0.48 (0.02) | **0.48 (0.02)** | 0.53 (0.03) | 0.49 (0.02) | 0.51 (0.02) | 0.51 (0.02) | 0.49 (0.02) | 0.50 (0.01) |
| | Transformer (rano) | 1 | 0.38 (0.01) | **0.35 (0.01)** | 0.35 (0.01) | 0.32 (0.01) | 0.36 (0.01) | 0.36 (0.01) | 0.37 (0.01) | 0.34 (0.01) |
| | Transformer (zale) | 1 | 0.44 (0.02) | **0.44 (0.01)** | 0.44 (0.03) | 0.49 (0.02) | 0.47 (0.01) | 0.46 (0.01) | 0.49 (0.01) | 0.47 (0.01) |
| **SMILES** | GRU | 0.05 | 2.47 (0.22) | **2.47 (0.22)** | 2.42 (0.21) | 2.24 (0.17) | 2.35 (0.23) | 2.42 (0.28) | 1.97 (0.18) | 2.25 (0.31) |
| | | 0.1 | 1.76 (0.3) | **2.57 (0.31)** | 2.45 (0.26) | 2.07 (0.34) | 2.24 (0.28) | 2.26 (0.33) | 2.24 (0.31) | 1.92 (0.24) |
| | | 1 | 2.43 (0.32) | **2.48 (0.29)** | 2.76 (0.24) | 2.17 (0.49) | 2.43 (0.32) | 2.35 (0.37) | 1.4 (0.52) | 2.04 (0.33) |
| | LSTM | 0.05 | 2.65 (0.32) | **2.73 (0.33)** | 2.89 (0.26) | 3.02 (0.34) | 2.64 (0.3) | 2.91 (0.29) | 2.77 (0.23) | 2.68 (0.37) |
| | | 0.1 | 1.97 (0.37) | **1.78 (0.43)** | 1.98 (0.4) | 2.16 (0.29) | 2.36 (0.41) | 1.87 (0.39) | 2.34 (0.33) | 1.73 (0.41) |
| | | 1 | 3.06 (0.2) | **2.71 (0.34)** | 2.65 (0.16) | 2.83 (0.25) | 2.75 (0.28) | 3.49 (0.26) | 2.68 (0.31) | 3.16 (0.35) |
| | Transformer | 0.05 | 2.47 (0.23) | **2.88 (0.24)** | 2.42 (0.17) | 2.67 (0.27) | 2.91 (0.3) | 2.25 (0.2) | 2.43 (0.24) | 2.48 (0.29) |
| | | 0.1 | 3.03 (0.3) | **2.15 (0.18)** | 2.51 (0.25) | 2.41 (0.24) | 2.28 (0.12) | 2.28 (0.22) | 2.53 (0.26) | 2.46 (0.16) |
| | | 1 | 2.37 (0.21) | **2.25 (0.18)** | 2.16 (0.17) | 2.21 (0.25) | 2.46 (0.2) | 2.11 (0.12) | 2.31 (0.16) | 2.71 (0.32) |

## B.2. LSO (L-BFGS) hyper-parameters

We provide an ablation study for the facet-length parameter. For each setup, the values reported in the main paper are highlighted in bold.

*Table 6.* Ablation study for the facet length parameter of LSO (L-BFGS) method. Average across 10 independent runs of the average top 20 values across datasets, architectures, and bound methods are displayed for facet lengths of size 1, 5 and 10. Results from the main paper are bold.

| | Architecture | $\beta$ | 1 | 5 | 10 |
|---|---|---|---|---|---|
| **Expressions** | GRU | 0.05 | -1.79 (0.08) | **-1.72 (0.07)** | -1.72 (0.07) |
| | | 0.1 | -1.73 (0.11) | **-1.93 (0.09)** | -1.89 (0.11) |
| | | 1 | -1.94 (0.07) | **-1.97 (0.07)** | -2.03 (0.09) |
| | LSTM | 0.05 | -1.89 (0.09) | **-1.78 (0.09)** | -1.93 (0.06) |
| | | 0.1 | -1.29 (0.06) | **-1.39 (0.08)** | -1.37 (0.06) |
| | | 1 | -2.04 (0.05) | **-2.04 (0.04)** | -2.04 (0.05) |
| | Transformer | 0.05 | -3.11 (0.14) | **-2.93 (0.13)** | -3.02 (0.09) |
| | | 0.1 | -3.19 (0.28) | **-2.69 (0.25)** | -2.93 (0.22) |
| | | 1 | -2.44 (0.11) | **-2.41 (0.09)** | -2.28 (0.11) |
| **SELFIES** | Transformer (pdop) | 0.05 | 0.22 (0.01) | **0.21 (0.01)** | 0.25 (0.01) |
| | | 0.1 | 0.25 (0.01) | **0.19 (0.01)** | 0.22 (0.03) |
| | | 1 | 0.26 (0.01) | **0.25 (0.01)** | 0.27 (0.01) |
| | Transformer (rano) | 0.05 | 0.04 (0.0) | **0.03 (0.0)** | – |
| | | 0.1 | – | **–** | – |
| | | 1 | 0.09 (0.02) | **0.07 (0.0)** | 0.08 (0.01) |
| | Transformer (zale) | 0.05 | 0.12 (0.02) | **0.13 (0.01)** | 0.11 (0.01) |
| | | 0.1 | 0.08 (0.01) | **0.06 (0.0)** | 0.07 (0.0) |
| | | 1 | 0.18 (0.02) | **0.18 (0.02)** | 0.19 (0.02) |
| **SELFIES (27)** | Transformer (pdop) | 1 | 0.45 (0.00) | **0.42 (0.00)** | 0.43 (0.01) |
| | Transformer (rano) | 1 | 0.22 (0.01) | **0.17 (0.01)** | 0.19 (0.01) |
| | Transformer (zale) | 1 | 0.4 (0.01) | **0.37 (0.01)** | 0.38 (0.01) |
| **SMILES** | GRU | 0.05 | 0.65 (0.12) | **0.52 (0.12)** | 0.52 (0.13) |
| | | 0.1 | 0.52 (0.14) | **0.12 (0.14)** | 0.36 (0.15) |
| | | 1 | – | **–** | – |
| | LSTM | 0.05 | 0.67 (0.15) | **0.66 (0.12)** | 0.56 (0.2) |
| | | 0.1 | – | **–** | – |
| | | 1 | 0.49 (0.16) | **0.7 (0.2)** | 0.55 (0.15) |
| | Transformer | 0.05 | 0.52 (0.18) | **0.42 (0.15)** | 0.55 (0.12) |
| | | 0.1 | 0.63 (0.18) | **0.61 (0.15)** | 0.56 (0.17) |
| | | 1 | 0.16 (0.15) | **0.23 (0.12)** | 0.39 (0.13) |

*Table 7.* Ablation study for the facet length parameter of LSO (L-BFGS) method. Average across 10 independent runs, of the best value across datasets, architectures, and bound methods are displayed for facet lengths of size 1, 5 and 10. Results from the main paper are bold.

| | Architecture | $\beta$ | 1 | 5 | 10 |
|---|---|---|---|---|---|
| **Expressions** | GRU | 0.05 | -0.57 (0.07) | **-0.56 (0.04)** | -0.56 (0.07) |
| | | 0.1 | -0.53 (0.04) | **-0.46 (0.02)** | -0.46 (0.02) |
| | | 1 | -0.68 (0.05) | **-0.77 (0.02)** | -0.73 (0.04) |
| | LSTM | 0.05 | -0.57 (0.11) | **-0.56 (0.04)** | -0.67 (0.09) |
| | | 0.1 | -0.4 (0.05) | **-0.44 (0.04)** | -0.47 (0.04) |
| | | 1 | -1.01 (0.1) | **-1.02 (0.07)** | -0.86 (0.0) |
| | Transformer | 0.05 | -1.06 (0.13) | **-0.8 (0.1)** | -1.11 (0.18) |
| | | 0.1 | -0.8 (0.14) | **-0.65 (0.1)** | -0.69 (0.09) |
| | | 1 | -0.85 (0.1) | **-0.82 (0.1)** | -0.78 (0.12) |
| **SELFIES** | Transformer (pdop) | 0.05 | 0.36 (0.01) | **0.36 (0.02)** | 0.34 (0.03) |
| | | 0.1 | 0.29 (0.03) | **0.34 (0.02)** | 0.27 (0.03) |
| | | 1 | 0.39 (0.01) | **0.36 (0.02)** | 0.35 (0.03) |
| | Transformer (rano) | 0.05 | 0.06 (0.01) | **0.07 (0.01)** | 0.08 (0.01) |
| | | 0.1 | 0.08 (0.02) | **0.08 (0.02)** | 0.07 (0.01) |
| | | 1 | 0.16 (0.02) | **0.16 (0.02)** | 0.16 (0.02) |
| | Transformer (zale) | 0.05 | 0.27 (0.03) | **0.27 (0.03)** | 0.26 (0.03) |
| | | 0.1 | 0.15 (0.04) | **0.19 (0.03)** | 0.18 (0.04) |
| | | 1 | 0.38 (0.01) | **0.38 (0.02)** | 0.39 (0.01) |
| **SELFIES 27** | Transformer (pdop) | 1 | 0.51 (0.02) | **0.47 (0.01)** | 0.48 (0.01) |
| | Transformer (rano) | 1 | 0.34 (0.02) | **0.37 (0.01)** | 0.34 (0.02) |
| | Transformer (zale) | 1 | 0.47 (0.0) | **0.44 (0.01)** | 0.45 (0.02) |
| **SMILES** | GRU | 0.05 | 2.06 (0.3) | **1.71 (0.24)** | 2.02 (0.22) |
| | | 0.1 | 2.11 (0.28) | **1.74 (0.2)** | 1.94 (0.22) |
| | | 1 | 2.4 (0.31) | **2.1 (0.32)** | 2.34 (0.31) |
| | LSTM | 0.05 | 2.31 (0.21) | **2.32 (0.19)** | 2.15 (0.18) |
| | | 0.1 | 1.74 (0.5) | **1.85 (0.34)** | 1.47 (0.49) |
| | | 1 | 2.8 (0.25) | **3.09 (0.16)** | 2.42 (0.3) |
| | Transformer | 0.05 | 1.82 (0.2) | **1.79 (0.13)** | 1.74 (0.17) |
| | | 0.1 | 2.19 (0.16) | **2.24 (0.1)** | 2.09 (0.17) |
| | | 1 | 1.72 (0.18) | **2.0 (0.14)** | 1.96 (0.14) |

## B.3. Likelihood hyper-parameters

We provide an ablation study on the effect of changing the regularization strength parameter $\lambda$. For each setup, the values reported in the main paper are highlighted in bold.

### B.3.1. EXPRESSIONS

*Table 8.* Ablation study for the regularization parameter $\lambda$ for Likelihood. We report the average of the top 20 solutions found for the Expressions VAEs.

| Architecture | $\beta$ | $\lambda = 0.05$ | $\lambda = 0.1$ | $\lambda = 0.2$ |
|---|---|---|---|---|
| | 0.05 | **-1.15** | -1.10 | -1.11 |
| GRU | 0.1 | **-1.31** | -1.16 | -1.14 |
| | 1 | **-0.88** | -1.04 | -1.15 |
| | 0.05 | **-1.00** | -1.05 | -1.17 |
| LSTM | 0.1 | **-0.75** | -0.81 | -0.72 |
| | 1 | **-1.81** | -1.79 | -1.78 |
| | 0.05 | **-0.93** | -0.95 | -0.98 |
| Transformer | 0.1 | **-0.81** | -0.74 | -0.70 |
| | 1 | **-1.45** | -1.42 | -1.19 |

*Table 9.* Ablation study for the regularization parameter $\lambda$ for Likelihood. We report the average of the top solution found for the Expressions VAEs.

| Architecture | $\beta$ | $\lambda = 0.05$ | $\lambda = 0.1$ | $\lambda = 0.2$ |
|---|---|---|---|---|
| | 0.05 | **-0.4** | -0.43 | -0.48 |
| GRU | 0.1 | **-0.43** | -0.42 | -0.47 |
| | 1 | **-0.43** | -0.58 | -0.59 |
| | 0.05 | **-0.4** | -0.45 | -0.44 |
| LSTM | 0.1 | **-0.32** | -0.39 | -0.36 |
| | 1 | **-0.86** | -0.86 | -0.86 |
| | 0.05 | **-0.38** | -0.39 | -0.43 |
| Transformer | 0.1 | **-0.41** | -0.39 | -0.39 |
| | 1 | **-0.65** | -0.50 | -0.42 |

B.3.2. SMILES

*Table 10.* Ablation study for the regularization parameter $\lambda$ for Likelihood. We report the average of the top 20 solutions found for the SMILES VAEs.

| Architecture | $\beta$ | $\lambda = 0.3$ | $\lambda = 0.5$ | $\lambda = 0.8$ |
|---|---|---|---|---|
| | 0.05 | 2.15 | **2.31** | 2.25 |
| GRU | 0.1 | 2.10 | **2.12** | 2.22 |
| | 1 | 0.60 | **1.49** | 1.40 |
| | 0.05 | 2.17 | **2.28** | 2.30 |
| LSTM | 0.1 | 0.81 | **1.43** | 1.58 |
| | 1 | 0.53 | **1.67** | 1.77 |
| | 0.05 | 2.43 | **2.25** | 2.32 |
| Transformer | 0.1 | 2.26 | **2.23** | 2.34 |
| | 1 | 2.01 | **2.16** | 1.98 |

*Table 11.* Ablation study for the regularization parameter $\lambda$ for Likelihood. We report the average of the top solution found for the SMILES VAEs.

| Architecture | $\beta$ | $\lambda = 0.3$ | $\lambda = 0.5$ | $\lambda = 0.8$ |
|---|---|---|---|---|
| | 0.05 | 2.96 | **3.26** | 3.27 |
| GRU | 0.1 | 3.02 | **3.33** | 3.07 |
| | 1 | 3.12 | **3.89** | 3.47 |
| | 0.05 | 3.30 | **3.37** | 3.19 |
| LSTM | 0.1 | 2.44 | **3.54** | 2.92 |
| | 1 | 3.00 | **3.48** | 2.94 |
| | 0.05 | 3.24 | **3.19** | 3.07 |
| Transformer | 0.1 | 3.10 | **3.16** | 3.25 |
| | 1 | 3.18 | **3.2** | 2.92 |

B.3.3. SELFIES

Table 12. Ablation study for the regularization parameter $\lambda$ for Likelihood. We report the average of the top 20 solutions found for the SELFIES VAEs.

| Objective | $\beta$ | $\lambda = 0.1$ | $\lambda = 0.3$ | $\lambda = 0.5$ |
|---|---|---|---|---|
| | 0.05 | **0.37** | 0.37 | 0.37 |
| Pdop | 0.1 | **0.36** | 0.36 | 0.34 |
| | 1 | **0.34** | 0.29 | 0.28 |
| | (Maus et al., 2022) | 0.42 | 0.40 | **0.35** |
| | 0.05 | **0.21** | 0.22 | 0.21 |
| Rano | 0.1 | **0.22** | 0.21 | 0.20 |
| | 1 | **0.21** | 0.23 | 0.19 |
| | (Maus et al., 2022) | **0.22** | 0.29 | 0.26 |
| | 0.05 | **0.33** | 0.33 | 0.32 |
| Zale | 0.1 | **0.34** | 0.33 | 0.32 |
| | 1 | **0.30** | 0.29 | 0.26 |
| | (Maus et al., 2022) | 0.41 | 0.39 | **0.31** |

Table 13. Ablation study for the regularization parameter $\lambda$ for Likelihood. We report the average of the top solution found for the SELFIES VAEs.

| Objective | $\beta$ | $\lambda = 0.1$ | $\lambda = 0.3$ | $\lambda = 0.5$ |
|---|---|---|---|---|
| | 0.05 | **0.43** | 0.43 | 0.42 |
| Pdop | 0.1 | **0.43** | 0.41 | 0.39 |
| | 1 | **0.40** | 0.40 | 0.37 |
| | (Maus et al., 2022) | 0.47 | 0.45 | **0.42** |
| | 0.05 | **0.31** | 0.31 | 0.29 |
| Rano | 0.1 | **0.33** | 0.30 | 0.30 |
| | 1 | **0.33** | 0.30 | 0.33 |
| | (Maus et al., 2022) | **0.34** | 0.36 | 0.37 |
| | 0.05 | **0.44** | 0.45 | 0.42 |
| Zale | 0.1 | **0.44** | 0.42 | 0.42 |
| | 1 | **0.42** | 0.40 | 0.37 |
| | (Maus et al., 2022) | 0.49 | 0.49 | **0.42** |

## B.4. LES hyper-parameters

We provide an ablation study on the effect of changing the regularization strength parameter $\lambda$. For each setup, the values reported in the main paper are highlighted in bold.

### B.4.1. EXPRESSIONS

*Table 14.* Ablation study for the regularization parameter $\lambda$ for LES. We report the average of the top 20 solutions found for the Expressions VAEs.

| **Architecture** | $\beta$ | $\lambda = 0.05$ | $\lambda = 0.1$ | $\lambda = 0.2$ |
|---|---|---|---|---|
| | 0.05 | **-1.43** | -1.28 | -1.32 |
| GRU | 0.1 | **-1.34** | -1.06 | -1.29 |
| | 1 | **-0.84** | -0.98 | -1.08 |
| | 0.05 | **-1.06** | -1.09 | -1.00 |
| LSTM | 0.1 | **-0.80** | -0.72 | -0.73 |
| | 1 | **-1.79** | -1.79 | -1.84 |
| | 0.05 | **-1.00** | -0.86 | -0.92 |
| Transformer | 0.1 | **-0.77** | -0.75 | -0.67 |
| | 1 | **-1.36** | -1.28 | -1.05 |

*Table 15.* Ablation study for the regularization parameter $\lambda$ for LES. We report the average of the top solution found for the Expressions VAEs.

| **Architecture** | $\beta$ | $\lambda = 0.05$ | $\lambda = 0.1$ | $\lambda = 0.2$ |
|---|---|---|---|---|
| | 0.05 | **-0.55** | -0.56 | -0.44 |
| GRU | 0.1 | **-0.45** | -0.44 | -0.41 |
| | 1 | **-0.47** | -0.47 | -0.52 |
| | 0.05 | **-0.43** | -0.42 | -0.41 |
| LSTM | 0.1 | **-0.32** | -0.39 | -0.39 |
| | 1 | **-0.86** | -0.86 | -0.86 |
| | 0.05 | **-0.43** | -0.39 | -0.41 |
| Transformer | 0.1 | **-0.36** | -0.39 | -0.32 |
| | 1 | **-0.52** | -0.52 | -0.45 |

### B.4.2. SMILES

*Table 16.* Ablation study for the regularization parameter $\lambda$ for LES. We report the average of the top 20 solutions found for the SMILES VAEs.

| **Architecture** | $\beta$ | $\lambda = 0.3$ | $\lambda = 0.5$ | $\lambda = 0.8$ |
|---|---|---|---|---|
| | 0.05 | 2.20 | **2.31** | 2.32 |
| GRU | 0.1 | 2.12 | **2.30** | 2.35 |
| | 1 | 0.38 | **1.64** | 1.77 |
| | 0.05 | 2.34 | **2.33** | 2.37 |
| LSTM | 0.1 | 0.94 | **1.57** | 1.80 |
| | 1 | 0.59 | **1.94** | 2.17 |
| | 0.05 | 2.30 | **2.26** | 2.35 |
| Transformer | 0.1 | 2.31 | **2.26** | 2.29 |
| | 1 | 2.09 | **2.17** | 2.30 |

*Table 17.* Ablation study for the regularization parameter $\lambda$ for LES. We report the average of the top solution found for the SMILES VAEs.

| Architecture | $\beta$ | $\lambda = 0.3$ | $\lambda = 0.5$ | $\lambda = 0.8$ |
|---|---|---|---|---|
| | 0.05 | 3.16 | **3.29** | 3.18 |
| GRU | 0.1 | 3.19 | **3.55** | 3.42 |
| | 1 | 3.13 | **3.85** | 3.53 |
| | 0.05 | 3.51 | **3.29** | 3.35 |
| LSTM | 0.1 | 3.10 | **3.54** | 2.93 |
| | 1 | 2.6 | **3.6** | 3.29 |
| | 0.05 | 3.16 | **3.21** | 3.35 |
| Transformer | 0.1 | 3.09 | **3.23** | 3.03 |
| | 1 | 2.94 | **3.2** | 3.17 |

### B.4.3. SELFIES

*Table 18.* Ablation study for the regularization parameter $\lambda$ for LES. We report the average of the top 20 solutions found for the SELFIES VAEs.

| Objective | $\beta$ | $\lambda = 0.1$ | $\lambda = 0.3$ | $\lambda = 0.5$ |
|---|---|---|---|---|
| | 0.05 | **0.37** | 0.37 | 0.37 |
| Pdop | 0.1 | **0.36** | 0.36 | 0.36 |
| | 1 | **0.35** | 0.32 | 0.33 |
| | (Maus et al., 2022) | 0.46 | 0.48 | **0.47** |
| | 0.05 | **0.21** | 0.22 | 0.21 |
| Rano | 0.1 | **0.21** | 0.21 | 0.19 |
| | 1 | **0.21** | 0.21 | 0.20 |
| | (Maus et al., 2022) | **0.30** | 0.29 | 0.30 |
| | 0.05 | **0.33** | 0.34 | 0.34 |
| Zale | 0.1 | **0.34** | 0.33 | 0.33 |
| | 1 | **0.31** | 0.31 | 0.31 |
| | (Maus et al., 2022) | 0.40 | 0.40 | **0.41** |

*Table 19.* Ablation study for the regularization parameter $\lambda$ for LES. We report the average of the top solution found for the SELFIES VAEs.

| **Objective** | $\beta$ | $\lambda = 0.1$ | $\lambda = 0.3$ | $\lambda = 0.5$ |
|---|---|---|---|---|
| Pdop | 0.05 | **0.43** | 0.43 | 0.42 |
| | 0.1 | **0.43** | 0.41 | 0.40 |
| | 1 | **0.41** | 0.42 | 0.40 |
| | (Maus et al., 2022) | 0.50 | 0.50 | **0.54** |
| Rano | 0.05 | **0.33** | 0.32 | 0.29 |
| | 0.1 | **0.33** | 0.30 | 0.31 |
| | 1 | **0.31** | 0.28 | 0.30 |
| | (Maus et al., 2022) | **0.37** | 0.37 | 0.39 |
| Zale | 0.05 | **0.43** | 0.45 | 0.45 |
| | 0.1 | **0.44** | 0.45 | 0.42 |
| | 1 | **0.42** | 0.43 | 0.39 |
| | (Maus et al., 2022) | 0.50 | 0.50 | **0.47** |

# C. Additional experimental results

*Table 20.* Average of the top solution found during LSO (higher is better), across datasets and decoder architectures. We **bold** the best method and underline the second-best. The average ranking for each method (lower is better) is provided, along with the number of times each method is within one standard deviation of the best. LES and Prior achieve the highest value most frequently (14 out of 30) and outperforms other methods both in terms of the average ranking and the frequency of being within one standard deviation of the best result.

| | Architecture | $\beta$ | LES | LSO (L-BFGS) | UC | LSO (GA) | Prior | TuRBO | Likelihood |
|---|---|---|---|---|---|---|---|---|---|
| **Expressions** | GRU | 0.05 | -0.55 (0.04) | -0.56 (0.04) | -0.59 (0.04) | -0.45 (0.05) | **-0.4 (0.07)** | -0.73 (0.12) | **-0.4 (0.07)** |
| | | 0.1 | -0.45 (0.03) | -0.46 (0.02) | -0.47 (0.05) | -0.43 (0.02) | **-0.37 (0.03)** | -0.61 (0.07) | -0.43 (0.02) |
| | | 1 | -0.47 (0.03) | -0.77 (0.02) | -0.51 (0.04) | -0.46 (0.02) | -0.47 (0.03) | -0.54 (0.05) | **-0.43 (0.01)** |
| | LSTM | 0.05 | -0.43 (0.02) | -0.56 (0.04) | -0.52 (0.05) | -0.43 (0.01) | -0.41 (0.01) | -0.43 (0.02) | **-0.4 (0.01)** |
| | | 0.1 | **-0.32 (0.05)** | -0.44 (0.04) | -0.39 (0.04) | -0.38 (0.02) | -0.4 (0.01) | -0.39 (0.0) | **-0.32 (0.04)** |
| | | 1 | **-0.86 (0.0)** | -1.02 (0.07) | **-0.86 (0.0)** | **-0.86 (0.0)** | **-0.86 (0.0)** | **-0.86 (0.0)** | -0.91 (0.04) |
| | Transformer | 0.05 | -0.43 (0.03) | -0.8 (0.1) | -0.57 (0.05) | -0.44 (0.02) | **-0.37 (0.05)** | -0.44 (0.02) | -0.38 (0.04) |
| | | 0.1 | -0.36 (0.03) | -0.65 (0.1) | -0.55 (0.04) | -0.39 (0.01) | **-0.35 (0.04)** | -0.41 (0.02) | -0.41 (0.02) |
| | | 1 | **-0.52 (0.05)** | -0.82 (0.1) | -0.58 (0.05) | -0.58 (0.04) | -0.62 (0.09) | -0.58 (0.1) | -0.65 (0.08) |
| **SELFIES** | Transformer (pdop) | 0.05 | **0.43 (0.0)** | 0.36 (0.02) | 0.42 (0.0) | 0.42 (0.0) | 0.42 (0.0) | 0.15 (0.03) | **0.43 (0.0)** |
| | | 0.1 | **0.43 (0.0)** | 0.34 (0.02) | 0.42 (0.01) | 0.42 (0.0) | 0.41 (0.01) | 0.09 (0.03) | **0.43 (0.01)** |
| | | 1 | **0.41 (0.01)** | 0.36 (0.02) | 0.39 (0.01) | 0.4 (0.0) | 0.38 (0.01) | 0.31 (0.03) | 0.4 (0.0) |
| | Transformer (rano) | 0.05 | 0.33 (0.01) | 0.07 (0.01) | **0.36 (0.01)** | 0.33 (0.01) | 0.32 (0.01) | 0.2 (0.02) | 0.31 (0.01) |
| | | 0.1 | 0.33 (0.01) | 0.08 (0.02) | **0.36 (0.02)** | 0.32 (0.01) | 0.32 (0.01) | 0.1 (0.01) | 0.33 (0.01) |
| | | 1 | 0.31 (0.01) | 0.16 (0.02) | **0.39 (0.02)** | 0.31 (0.01) | 0.33 (0.02) | 0.05 (0.02) | 0.33 (0.01) |
| | Transformer (zale) | 0.05 | 0.43 (0.01) | 0.27 (0.03) | 0.42 (0.01) | 0.43 (0.01) | 0.42 (0.0) | 0.15 (0.03) | **0.44 (0.01)** |
| | | 0.1 | **0.44 (0.01)** | 0.19 (0.03) | 0.42 (0.01) | 0.42 (0.01) | 0.42 (0.01) | 0.16 (0.01) | **0.44 (0.01)** |
| | | 1 | **0.42 (0.01)** | 0.38 (0.02) | 0.39 (0.01) | **0.42 (0.01)** | 0.37 (0.01) | 0.31 (0.03) | **0.42 (0.01)** |
| **SELFIES (27)** | Transformer (pdop) | 1 | **0.54 (0.01)** | 0.46 (0.01) | 0.50 (0.01) | 0.49 (0.01) | 0.36 (0.02) | 0.48 (0.02) | 0.42 (0.00) |
| | Transformer (rano) | 1 | **0.37 (0.00)** | **0.37 (0.01)** | 0.36 (0.01) | **0.37 (0.01)** | 0.34 (0.01) | 0.35 (0.02) | 0.34 (0.01) |
| | Transformer (zale) | 1 | **0.47 (0.01)** | 0.43 (0.01) | 0.44 (0.01) | **0.47 (0.01)** | 0.39 (0.01) | 0.44 (0.01) | 0.42 (0.01) |
| **SMILES** | GRU | 0.05 | **3.29 (0.1)** | 1.71 (0.24) | 3.13 (0.07) | 3.26 (0.11) | 3.18 (0.06) | 2.47 (0.22) | 3.26 (0.08) |
| | | 0.1 | **3.55 (0.14)** | 1.74 (0.2) | 3.2 (0.1) | 3.31 (0.16) | 3.15 (0.11) | 2.57 (0.31) | 3.33 (0.12) |
| | | 1 | 3.85 (0.17) | 2.1 (0.32) | 2.24 (0.28) | 3.66 (0.16) | **3.89 (0.28)** | 2.48 (0.29) | **3.89 (0.2)** |
| | LSTM | 0.05 | 3.29 (0.07) | 2.32 (0.19) | 3.12 (0.08) | 3.3 (0.1) | 3.28 (0.1) | 2.73 (0.33) | **3.37 (0.09)** |
| | | 0.1 | **3.66 (0.2)** | 1.85 (0.34) | 2.65 (0.19) | 3.52 (0.22) | 3.57 (0.18) | 1.78 (0.43) | 3.54 (0.16) |
| | | 1 | **3.6 (0.14)** | 3.09 (0.16) | 2.6 (0.3) | 3.18 (0.17) | 3.28 (0.17) | 2.71 (0.34) | 3.48 (0.11) |
| | Transformer | 0.05 | **3.21 (0.08)** | 1.79 (0.13) | 3.1 (0.07) | 3.14 (0.04) | 3.14 (0.04) | 2.88 (0.24) | 3.19 (0.08) |
| | | 0.1 | 3.23 (0.04) | 2.24 (0.1) | **3.28 (0.08)** | 3.11 (0.05) | 3.09 (0.06) | 2.15 (0.18) | 3.16 (0.06) |
| | | 1 | **3.2 (0.07)** | 2.0 (0.14) | 2.8 (0.13) | 3.13 (0.07) | 3.11 (0.1) | 2.25 (0.18) | **3.2 (0.06)** |
| | **Average rank** | | **2.15** | 6.17 | 4.17 | 3.15 | 3.86 | 5.7 | 2.76 |
| | **# within 1 std of best** | | **19** | 1 | 5 | 8 | 10 | 2 | 16 |

*Table 21.* Average of the top 20 solutions found during LSO (higher is better) across datasets and decoder architectures. We **bold** the best method and underline the second-best. The average ranking for each method (lower is better) is provided, along with the number of times each method is within one standard deviation of the best. LES achieves the highest value in most experiments (20 out of 30) and outperforms other methods in terms of both the average ranking and the frequency of being within one standard deviation of the best result.

| | Architecture | $\beta$ | LES | LSO (L-BFGS) | UC | LSO (GA) | Prior | TuRBO | Likelihood |
|---|---|---|---|---|---|---|---|---|---|
| **Expressions** | GRU | 0.05 | -1.43 (0.05) | -1.72 (0.07) | -1.5 (0.05) | -1.25 (0.06) | -1.28 (0.05) | -2.1 (0.11) | **-1.15 (0.09)** |
| | | 0.1 | -1.34 (0.08) | -1.93 (0.09) | -2.01 (0.12) | -1.21 (0.09) | **-1.18 (0.11)** | -2.14 (0.18) | -1.31 (0.08) |
| | | 1 | **-0.84 (0.02)** | -1.97 (0.07) | -0.91 (0.03) | **-0.84 (0.02)** | -0.89 (0.05) | -1.34 (0.05) | -0.88 (0.03) |
| | LSTM | 0.05 | -1.06 (0.06) | -1.78 (0.09) | -1.53 (0.06) | **-0.98 (0.05)** | -1.02 (0.03) | -1.59 (0.13) | -1.0 (0.07) |
| | | 0.1 | -0.8 (0.03) | -1.39 (0.08) | -1.09 (0.04) | -0.79 (0.03) | -0.79 (0.03) | -1.32 (0.13) | **-0.75 (0.03)** |
| | | 1 | **-1.79 (0.02)** | -2.04 (0.04) | -2.02 (0.03) | -1.81 (0.02) | -1.83 (0.02) | -2.22 (0.08) | -1.81 (0.03) |
| | Transformer | 0.05 | -1.0 (0.04) | -2.93 (0.13) | -1.64 (0.08) | -1.03 (0.04) | -0.99 (0.05) | -2.16 (0.09) | **-0.93 (0.05)** |
| | | 0.1 | **-0.77 (0.02)** | -2.69 (0.25) | -1.79 (0.08) | **-0.77 (0.04)** | -0.78 (0.03) | -1.39 (0.12) | -0.81 (0.03) |
| | | 1 | **-1.36 (0.09)** | -2.41 (0.09) | -1.52 (0.09) | -1.5 (0.07) | -1.41 (0.08) | -1.77 (0.12) | -1.45 (0.12) |
| **SELFIES** | Transformer (pdop) | 0.05 | **0.37 (0.0)** | 0.21 (0.01) | 0.36 (0.0) | **0.37 (0.0)** | **0.37 (0.0)** | 0.04 (0.01) | **0.37 (0.0)** |
| | | 0.1 | **0.36 (0.0)** | 0.19 (0.01) | 0.35 (0.0) | **0.36 (0.0)** | 0.34 (0.0) | 0.01 (0.0) | **0.36 (0.0)** |
| | | 1 | **0.35 (0.0)** | 0.25 (0.01) | 0.31 (0.01) | 0.33 (0.0) | 0.28 (0.01) | 0.23 (0.02) | 0.34 (0.0) |
| | Transformer (rano) | 0.05 | 0.21 (0.0) | 0.03 (0.0) | **0.22 (0.0)** | 0.21 (0.0) | 0.21 (0.0) | 0.06 (0.0) | 0.21 (0.0) |
| | | 0.1 | 0.21 (0.0) | – | **0.23 (0.01)** | 0.21 (0.0) | 0.22 (0.0) | 0.03 (0.01) | 0.22 (0.0) |
| | | 1 | 0.21 (0.0) | 0.07 (0.0) | **0.22 (0.01)** | 0.19 (0.0) | 0.2 (0.0) | – | 0.21 (0.0) |
| | Transformer (zale) | 0.05 | **0.33 (0.0)** | 0.13 (0.01) | 0.32 (0.0) | **0.33 (0.0)** | 0.32 (0.0) | 0.04 (0.01) | **0.33 (0.0)** |
| | | 0.1 | **0.34 (0.0)** | 0.06 (0.0) | 0.31 (0.01) | 0.32 (0.01) | 0.31 (0.0) | 0.04 (0.01) | **0.34 (0.0)** |
| | | 1 | **0.31 (0.0)** | 0.18 (0.02) | 0.26 (0.01) | 0.29 (0.01) | 0.23 (0.01) | 0.17 (0.02) | 0.3 (0.01) |
| **SELFIES (27)** | Transformer (pdop) | 1 | **0.48 (0.01)** | 0.42 (0.01) | 0.44 (0.00) | 0.44 (0.00) | 0.3 (0.01) | 0.44 (0.01) | 0.35 (0.01) |
| | Transformer (rano) | 1 | **0.29 (0.00)** | 0.17 (0.03) | 0.27 (0.01) | 0.27 (0.01) | **0.25 (0.01)** | 0.22 (0.03) | 0.22 (0.01) |
| | Transformer (zale) | 1 | **0.4 (0.00)** | 0.37 (0.01) | 0.38 (0.01) | 0.39 (0.0) | 0.27 (0.01) | 0.37 (0.01) | 0.31 (0.01) |
| **SMILES** | GRU | 0.05 | **2.31 (0.04)** | 0.52 (0.12) | 2.18 (0.05) | 2.21 (0.05) | 2.2 (0.04) | 0.82 (0.15) | **2.31 (0.03)** |
| | | 0.1 | **2.3 (0.05)** | 0.12 (0.14) | 1.68 (0.04) | 2.09 (0.07) | 1.93 (0.06) | 0.41 (0.19) | 2.12 (0.07) |
| | | 1 | **1.64 (0.14)** | – | – | 1.19 (0.2) | 0.58 (0.28) | -0.16 (0.0) | 1.49 (0.14) |
| | LSTM | 0.05 | **2.33 (0.03)** | 0.66 (0.12) | 2.02 (0.03) | 2.22 (0.05) | 2.13 (0.05) | 1.54 (0.41) | 2.28 (0.03) |
| | | 0.1 | **1.57 (0.1)** | – | – | 0.8 (0.12) | 0.9 (0.09) | – | 1.43 (0.12) |
| | | 1 | **1.94 (0.1)** | 0.7 (0.2) | 0.95 (0.24) | 1.14 (0.21) | 0.81 (0.26) | 0.52 (0.26) | 1.67 (0.13) |
| | Transformer | 0.05 | **2.26 (0.04)** | 0.42 (0.15) | 2.04 (0.05) | 2.22 (0.03) | 2.24 (0.03) | 0.97 (0.19) | 2.25 (0.03) |
| | | 0.1 | **2.26 (0.03)** | 0.61 (0.15) | 2.08 (0.04) | 2.21 (0.03) | 2.17 (0.03) | 0.62 (0.14) | 2.23 (0.02) |
| | | 1 | **2.17 (0.05)** | 0.23 (0.12) | 1.32 (0.09) | 1.98 (0.06) | 1.82 (0.06) | 0.48 (0.18) | 2.16 (0.05) |
| | **Average rank** | | **1.9** | 6.4 | 4.3 | 2.87 | 3.8 | 6.1 | 2.53 |
| | **# within 1 std of best** | | **22** | 0 | 3 | 7 | 5 | 0 | 13 |

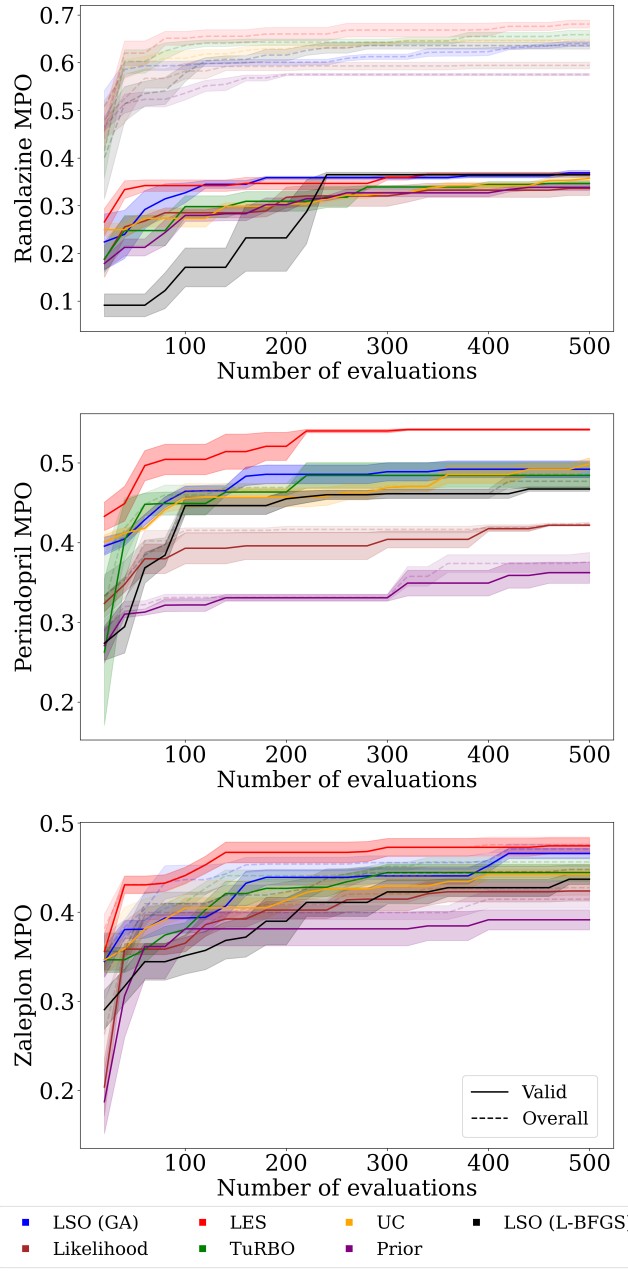

*Figure 4.* Cumulative true objective for the best solution found during Bayesian optimization with the pre-trained SELFIES-VAE (27). Each method is shown in a distinct color. Solid lines represent solutions passing quality filters, while dashed lines include all evaluations. LES achieves the best performance on Perindopril MPO and is comptetitive on Zaleplon MPO and Ranolazine MPO.

*Table 22.* AUROC values (higher is better) for identifying valid data points within the latent space, across datasets and decoder architectures. Data points are sampled from the training data, the VAE prior (standard Gaussian) and out of distribution (Gaussian with std of 5). LES achieves the best performance in most cases (18 out of 22) and on average. In addition, LES achieves AUROC values of at least 0.75 in all cases, indicating it can effectively differentiate valid from invalid data points.

| Dataset | Arch. | $\beta$ | LES | Polarity | Prior | Uncertainty | Train distances | Likelihood |
|---|---|---|---|---|---|---|---|---|
| SMILES | GRU | 0.05 | 0.93 | 0.42 | 0.09 | 0.85 | 0.85 | **0.94** |
| | | 0.1 | **0.94** | 0.72 | 0.12 | 0.84 | 0.86 | **0.94** |
| | | 1.0 | 0.91 | 0.35 | 0.16 | 0.87 | 0.80 | **0.92** |
| | LSTM | 0.05 | **0.99** | 0.93 | 0.07 | **0.99** | 0.91 | 0.98 |
| | | 0.1 | 0.89 | 0.67 | 0.21 | 0.89 | 0.81 | **0.9** |
| | | 1.0 | **0.97** | 0.76 | 0.12 | 0.95 | 0.85 | 0.96 |
| | Transformer | 0.05 | **0.93** | 0.89 | 0.14 | 0.84 | 0.77 | 0.92 |
| | | 0.1 | **0.94** | 0.91 | 0.14 | 0.87 | 0.86 | 0.93 |
| | | 1.0 | **0.97** | 0.93 | 0.10 | 0.89 | 0.85 | 0.95 |
| Expressions | GRU | 0.05 | **0.96** | 0.89 | 0.38 | **0.96** | 0.67 | 0.88 |
| | | 0.1 | **0.94** | 0.86 | 0.42 | **0.94** | 0.71 | 0.8 |
| | | 1.0 | **0.94** | 0.80 | 0.57 | **0.94** | 0.75 | 0.89 |
| | LSTM | 0.05 | **0.96** | 0.86 | 0.38 | 0.90 | 0.67 | **0.96** |
| | | 0.1 | **0.95** | 0.83 | 0.37 | 0.91 | 0.66 | **0.95** |
| | | 1.0 | **0.95** | 0.79 | 0.56 | 0.91 | 0.72 | 0.91 |
| | Transformer | 0.05 | **0.91** | 0.79 | 0.43 | 0.86 | 0.71 | 0.90 |
| | | 0.1 | **0.91** | 0.79 | 0.53 | 0.87 | 0.78 | 0.89 |
| | | 1.0 | 0.86 | 0.61 | 0.70 | **0.92** | 0.89 | **0.92** |
| SELFIES | Transformer | 0.05 | **1.0** | 0.99 | 0.02 | 0.99 | 0.62 | 0.97 |
| | | 0.1 | **0.99** | 0.98 | 0.03 | 0.96 | 0.81 | 0.98 |
| | | 1.0 | **0.95** | 0.94 | 0.06 | 0.85 | 0.72 | 0.93 |
| SELFIES (27) | Transformer | – | **0.75** | 0.39 | 0.69 | 0.33 | 0.69 | 0.70 |
| | Average | | **0.93** | 0.78 | 0.29 | 0.88 | 0.77 | 0.91 |

*Table 23.* Effect of increasing $\lambda$ parameter for LES, for the expressions dataset. Increasing the value of the parameter $\lambda$ increases the percentage of valid solution in all cases.

| Architecture | $\beta$ | **LES** ($\lambda = 0.05$) | **LES** ($\lambda = 0.5$) |
|---|---|---|---|
| | **0.05** | 0.92 | 0.96 |
| GRU | **0.1** | 0.7 | 0.82 |
| | **1** | 0.72 | 0.87 |
| | **0.05** | 0.93 | 0.98 |
| LSTM | **0.1** | 0.93 | 0.98 |
| | **1** | 0.76 | 0.97 |
| | **0.05** | 0.87 | 0.94 |
| Transformer | **0.1** | 0.88 | 0.96 |
| | **1** | 0.83 | 0.85 |

## D. Background on related work

**Bayesian uncertainty (Notin et al., 2021)**   Under a Bayesian viewpoint, the trained neural network parameters ($\boldsymbol{\theta}$) follow a variational distribution, which we can sample from using MC-Dropout (Gal & Ghahramani, 2016). Based on this distribution, the uncertainty is defined as

$$\mathcal{M}(\boldsymbol{z}) = \mathcal{H}_p(p(\boldsymbol{X}|\boldsymbol{Z} = \boldsymbol{z})) - \mathbb{E}_{\boldsymbol{\theta}}\mathcal{H}_{p_{\boldsymbol{\theta}}}(p_{\boldsymbol{\theta}}(\boldsymbol{X}|\boldsymbol{Z} = \boldsymbol{z})), \tag{36}$$

where $\mathcal{H}$ is the entropy and $p(\boldsymbol{X}|\boldsymbol{Z} = \boldsymbol{z})$ is the posterior predictive distribution. The uncertainty is estimated using MC-Dropout with important sampling (using the posteior predictive as the importance distribution) designed to approximate the expectations over the random variable $\boldsymbol{X}$, as it is typically a very large space (i.e., the sample of molecules that can implemented using a SMILE string with 120 characters)

*Table 24.* Proportion of valid solutions found during LSO (higher is better) across datasets and decoder architectures.We bold the best method (higher is better) and underline the second best. LES improves the validity of the solutions compared with LSO (GA) (which is LES with $\lambda = 0$) across all datasets.

| | Architecture | $\beta$ | LES | LSO (L-BFGS) | UC | LSO (GA) | Prior | TuRBO | Likelihood |
|---|---|---|---|---|---|---|---|---|---|
| **Expressions** | GRU | 0.05 | 0.92 | 0.59 | **1.0** | 0.91 | 0.91 | 0.94 | 0.89 |
| | | 0.1 | 0.7 | 0.57 | **1.0** | 0.64 | 0.66 | 0.9 | 0.67 |
| | | 1 | 0.72 | 0.45 | **0.99** | 0.69 | 0.69 | 0.83 | 0.69 |
| | LSTM | 0.05 | 0.93 | 0.62 | **1.0** | 0.88 | 0.89 | 0.94 | 0.92 |
| | | 0.1 | 0.93 | 0.67 | **1.0** | 0.9 | 0.89 | 0.94 | 0.92 |
| | | 1 | 0.76 | 0.58 | **0.99** | 0.65 | 0.66 | 0.89 | 0.73 |
| | Transformer | 0.05 | 0.87 | 0.37 | **1.0** | 0.83 | 0.85 | 0.9 | 0.84 |
| | | 0.1 | 0.88 | 0.28 | **1.0** | 0.8 | 0.81 | 0.89 | 0.85 |
| | | 1 | 0.83 | 0.36 | **1.0** | 0.74 | 0.76 | 0.87 | 0.79 |
| **SELFIES** | Transformer (pdop) | 0.05 | 0.8 | 0.05 | **0.81** | 0.76 | 0.73 | 0.14 | 0.77 |
| | | 0.1 | 0.68 | 0.04 | **0.71** | 0.62 | 0.56 | 0.14 | 0.64 |
| | | 1 | **0.59** | 0.08 | 0.55 | 0.49 | 0.44 | 0.08 | 0.53 |
| | Transformer (rano) | 0.05 | 0.66 | 0.02 | **0.73** | 0.61 | 0.57 | 0.09 | 0.62 |
| | | 0.1 | 0.57 | 0.01 | **0.67** | 0.52 | 0.44 | 0.04 | 0.53 |
| | | 1 | 0.43 | 0.02 | **0.49** | 0.36 | 0.28 | 0.01 | 0.39 |
| | Transformer (zale) | 0.05 | 0.71 | 0.08 | **0.74** | 0.66 | 0.62 | 0.12 | 0.69 |
| | | 0.1 | 0.66 | 0.01 | **0.67** | 0.59 | 0.51 | 0.08 | 0.63 |
| | | 1 | **0.54** | 0.18 | 0.47 | 0.43 | 0.35 | 0.17 | 0.46 |
| **SELFIES (27)** | Transformer (pdop) | 1 | 0.69 | 0.45 | 0.56 | 0.58 | **0.75** | 0.43 | 0.68 |
| | Transformer (rano) | 1 | **0.26** | 0.07 | 0.14 | 0.19 | 0.20 | 0.11 | 0.15 |
| | Transformer (zale) | 1 | 0.65 | 0.52 | 0.66 | 0.66 | 0.75 | 0.48 | **0.76** |
| **SMILES** | GRU | 0.05 | **0.61** | 0.14 | 0.48 | 0.47 | 0.44 | 0.12 | 0.59 |
| | | 0.1 | **0.37** | 0.07 | 0.25 | 0.23 | 0.22 | 0.06 | 0.35 |
| | | 1 | **0.09** | 0.02 | 0.05 | 0.05 | 0.04 | 0.02 | 0.08 |
| | LSTM | 0.05 | **0.6** | 0.11 | 0.44 | 0.42 | 0.39 | 0.11 | 0.57 |
| | | 0.1 | **0.08** | 0.01 | 0.06 | 0.05 | 0.04 | 0.01 | 0.07 |
| | | 1 | **0.16** | 0.09 | 0.07 | 0.08 | 0.07 | 0.09 | 0.12 |
| | Transformer | 0.05 | 0.7 | 0.42 | 0.7 | 0.68 | 0.68 | 0.31 | **0.72** |
| | | 0.1 | 0.65 | **0.67** | 0.63 | 0.6 | 0.55 | 0.41 | 0.64 |
| | | 1 | **0.48** | 0.35 | 0.46 | 0.34 | 0.26 | 0.3 | 0.45 |
| | **Average** | | 0.62 | 0.26 | **0.64** | 0.55 | 0.53 | 0.38 | 0.59 |

