# OpenReview forum: "MITIGATING OVER-EXPLORATION IN LATENT SPACE OPTIMIZATION USING LES"
_ICML.cc/2025/Conference — ICML 2025 poster_

### Official Review · Reviewer_oSTV · 2025-03-07

**Overall Recommendation:** 3

**Summary:**

This paper proposes a new score for optimizing the encoded latent space from pre-trained VAEs. The proposed new score employs the pre-trained decoder as a constraint to mitigate the over-exploration problem.

**Claims And Evidence:**

The author is motivated to use a pre-trained decoder (Sec.3, why use the sequence density function)  and provides empirical validation for the motivation. With my limited experience in this specific area, I believe no theoretical evidence is made. As such, I think it would be better to further validate the motivation.

**Essential References Not Discussed:**

Due to my limited knowledge, I think the author conducted extensive experiments to demonstrate and discuss the model for baselines.

**Experimental Designs Or Analyses:**

1. Despite most parts beyond my expertise, why does Eqn.13 compute likelihood for only p(x|z) instead of the log-likelihood over p(x)?

2. By comparing the performance of likelihood and LSO, it seems Likelihood can render competitive performance (sometimes better) while requiring much less computational cost. How does the author address such a balance?

**Methods And Evaluation Criteria:**

The author conducts various evaluations based on standard benchmarks and performs ablation studies for different VAE backbones.

**Other Comments Or Suggestions:**

The paper is, in general, well-written, while it can be organized better (e.g., the three examples in the introduction can be in the background).

**Other Strengths And Weaknesses:**

N/A

**Questions For Authors:**

n/a

**Relation To Broader Scientific Literature:**

The proposed idea is to address the problem of over-exploration issues in prior LSO methods.

**Theoretical Claims:**

N/A.

---

> ### Author Rebuttal · Authors · 2025-04-01
>
> We thank the reviewer for their time and effort in reviewing our work.
>
> ### Experimental Design and Analysis
>
> We calculate Equation 13 for a given **Z** because we are interested in a score that can be attributed to a single latent vector, rather than to a distribution over the output space \( p(x) \). Indeed, \( p(x) \) is not a function of a single latent vector and cannot be used to guide the optimization. We will make sure to emphasize this point in the revised version of our manuscript, under Equation 13.
>
> ** likelihood vs LES **
>
> The reviewer is making a great point, and we are glad to see that the reviewer agrees with us (lines 234–237) that:
>
> > “As the Likelihood score requires only a single forward pass of the decoder, it offers a more computationally efficient alternative, which comes with some performance trade-offs.”
>
> It is important for us to provide the full picture so that practitioners can use LES in cases where they feel the juice is worth the squeeze, but can also opt for the likelihood score if they have a limited computational budget.

---

### Official Review · Reviewer_A6Zc · 2025-03-09

**Overall Recommendation:** 4

**Summary:**

The paper develops Latent Exploration Score (LES) to mitigate the over-exporation in Latent Space Optimization (LSO). LES can be applied to any VAE decoder, and the paper develops a numerical procedure to incorporate LES as a constraint in LSO. The paper evaluates LES on multiple benchmark tests and demonstrates that LES improves the existing results.

**Claims And Evidence:**

Yes.

**Essential References Not Discussed:**

not I'm aware of.

**Experimental Designs Or Analyses:**

Yes. The analyses look sound.

**Methods And Evaluation Criteria:**

Yes, the proposed methods and evaluation on multiple benchmarks based on AUROC values make sense to me.

**Other Comments Or Suggestions:**

1. it might be better to add a link to the appendix D Bayesian uncertainty score on line 220 (LES is compared with the Bayesian uncertainty score...)?

2. Table 3 on line 367-368 is linked to wrong place.

**Other Strengths And Weaknesses:**

The paper is clearly-written, theoretically sound, and has novel idea. The experiment results also look good. I just have few questions/comments below.

**Questions For Authors:**

1. would the code used for experiments in this paper be released if the paper is accepted?
2. could you provide some insights on why UC finds more valid solutions (especially for expressions task) in table 13, but this doesn't seem to transfer to better AUROC values?
3. could you provide some guidelines on how one should tune the λ parameter in practice?

**Relation To Broader Scientific Literature:**

The paper is related to the broader literature on VAE, latent space optimization, and the over-exploitation issue. The paper has discussed it in details in the background section.

**Theoretical Claims:**

Yes, the proof looks correct to me.

---

> ### Author Rebuttal · Authors · 2025-04-01
>
> We thank the reviewer for their time and effort in reviewing our work.
>
> ### Comments and Suggestions
>
> We thank the reviewer for their comments and suggestions, which we will incorporate into the camera-ready version of our work.
>
> ### Question
>
> **All code and trained VAEs will be released if the paper is accepted.**
>
> We believe the UC early stopping procedure is too conservative, and essentially gets stuck around the initial data point—especially if the landscape is not smooth in terms of valid and invalid regions. Figure 1 provides some evidence that this is the case for the expression dataset. We will add this explanation in our revised version (lines 400–405) for further clarity.
>
> In practice, we believe that for the more interesting molecular tasks, $\lambda$ = 0.5 should generally perform better than what is currently used (e.g., Turbo). We provide this guideline in lines 370–372:
>
> > “Based on our findings, we suggest $\lambda$ = 0.5 as a reasonable default value, while leaving the exploration of an optimal choice for future work.”

---

### Official Review · Reviewer_p6GH · 2025-03-12

**Overall Recommendation:** 3

**Summary:**

This work targets the problem of over exploration in latent space optimization. The overarching problem is that when we optimize over discrete data in the latent space of a generative model e.g. a VAE, we often select points that decode poorly or are otherwise invalid. This work proposes a score (LES) to guide the latent space optimization process as a constraint. LES is informed by the decoder loss in order to score points higher when they're closer to the training data. Formally, LES is defined to be the output log-density of the decoder

This method is evaluated on more than 20 VAE models over 5 problem domains such as SMILES, SELFIES, expressions, etc.
The LES score is evaluated against baselines such as a "max probability" score, an uncertainty-based approach, the density of the prior, the polarity score, and also a score based on distance to training data.

----------------------------
### update after rebuttal

I appreciate the additional clarifications and experiments. I will keep my (positive) score then same.

**Claims And Evidence:**

Claims:
1. The LES score increases the number of valid solutions when used as a constraint in LSO
- This is quantified in Figure 3 and Table 3, where LES produces the highest percentage of valid solutions
2. LES is 80% faster than alternative approaches
- This is not well supported by Table 1. It looks like the likelihood score is 5-10X faster; the likelihood score also performs reasonably well on OOD data. It is worth nothing that the likelihood score underperforms in real tasks.
3. LES-constrained DSO performs better than six baseline approaches.
- Given the BO and OOD experiments, (tables 2 and 3 respectively), this is well supported by the experimental evidence.

**Essential References Not Discussed:**

I did not find any additional references that need to be included.

**Experimental Designs Or Analyses:**

I checked the following experiments:
* AUROC of OOD data shown in Table 2
- The setup for OOD data seems reasonable, if not a little contrived. Part of the motivation for this work was that other approaches find solutions that are invalid under the decoder. Is a Gaussian with std=5 enough to map to this scenario?
* Score computation time shown in Table 1
- No issues with timing, though likelihood is quite fast for the amount of performance that you lose.
* Bayesian Optimization experiments shown in Figure / Table 3.
- I don't have issues with the BO setup.

**Methods And Evaluation Criteria:**

I checked the benchmark datasets e.g. SMILES, expressions, SELFIES.
These datasets all feature discrete optimization settings that are reasonable for evaluating and using a score like LES.

**Other Comments Or Suggestions:**

N/A

**Other Strengths And Weaknesses:**

* I appreciate that this method is "decoder aware" given that any sample has to plausibly map back to a training-like sample under the decoder, the proposed method ensures that we're much more likely to get a valid sample -- which is what we really care about in applications.

* I found this paper to be well-written.

**Questions For Authors:**

N/A

**Relation To Broader Scientific Literature:**

This work found an appropriate hole in the research landscape. Namely, that many samples returned by black-box optimization in the latent space of deep generative models do not decode into valid samples. Other methods that tackle this problem are either simplistic (sample from the prior) or else use the decoder in a round-about way such as using the uncertainty of the decoder to score samples [1]. This method sits in between the two, by using the log-determinant of the decoder density directly we can see if a sample is valid.

[1] Notin, Pascal, José Miguel Hernández-Lobato, and Yarin Gal. "Improving black-box optimization in VAE latent space using decoder uncertainty." Advances in Neural Information Processing Systems 34 (2021): 802-814.

**Theoretical Claims:**

I checked the proof of lemma / theorem 3.1, I did not find any issues.

---

> ### Author Rebuttal · Authors · 2025-04-01
>
> We thank the reviewer for their time and effort in reviewing our work. We are glad to see that the reviewer finds our work to address an appropriate hole in the research landscape.
>
> We would like to clarify that we did not claim that LES is faster than all alternative approaches, but rather that it is faster than Bayesian uncertainty methods (lines 231–236):
>
> > “LES is computed faster than the Uncertainty score, achieving reductions of up to 85% in some cases. As the Likelihood score requires only a single forward pass of the decoder, it offers a more computationally efficient alternative, which comes with some performance trade-offs.”
>
> We believe that this claim is well-supported by Table 1.
>
> ### Is Gaussian with std = 5 OOD?
>
> We believe that a standard deviation of 5 is sufficient to qualify as out-of-distribution (OOD). To substantiate this claim, we provide the proportion of valid solutions for each VAE on a random sample of 500 data points generated with std = 5:
>
> | Architecture | Beta  | % Valid | Dataset     |
> |--------------|-------|---------|-------------|
> | Transformer  | 0.05  | 0.00    | SELFIES     |
> | Transformer  | 0.1   | 0.01    | SELFIES     |
> | Transformer  | 1     | 0.14    | SELFIES     |
> | Pre-trained  | —     | 0.09    | SELFIES     |
> | GRU          | 0.05  | 0.07    | SMILES      |
> | GRU          | 0.1   | 0.01    | SMILES      |
> | GRU          | 1     | 0.00    | SMILES      |
> | LSTM         | 0.05  | 0.00    | SMILES      |
> | LSTM         | 0.1   | 0.01    | SMILES      |
> | LSTM         | 1     | 0.00    | SMILES      |
> | Transformer  | 0.05  | 0.02    | SMILES      |
> | Transformer  | 0.1   | 0.03    | SMILES      |
> | Transformer  | 1     | 0.00    | SMILES      |
> | GRU          | 0.05  | 0.79    | Expressions |
> | GRU          | 0.1   | 0.67    | Expressions |
> | GRU          | 1     | 0.61    | Expressions |
> | LSTM         | 0.05  | 0.81    | Expressions |
> | LSTM         | 0.1   | 0.80    | Expressions |
> | LSTM         | 1     | 0.64    | Expressions |
> | Transformer  | 0.05  | 0.67    | Expressions |
> | Transformer  | 0.1   | 0.46    | Expressions |
> | Transformer  | 1     | 0.29    | Expressions |

---

### Official Review · Reviewer_BNj6 · 2025-03-15

**Overall Recommendation:** 2

**Summary:**

The paper proposes a latent exploration score method to mitigate over-exploration in latent space optimization, utilizing VAEs' properties for continuous optimization. It also demonstrates cases where solutions may become impractical.

**Claims And Evidence:**

The paper begins by examining cases of over-exploration in latent space optimization and proposes a method to efficiently identify regions that preserve the structure of the sequence space. The authors assert that all decoder neural networks in their study can be accurately approximated as continuous piecewise affine functions.

**Essential References Not Discussed:**

Given that this analysis focuses on latent space dynamics, would it be valuable to include additional visualizations correlating latent space movement with variations in generated outputs?

**Experimental Designs Or Analyses:**

The authors experimentally validated the relationship between the Latent Exploration Score (LES) and valid generation through multiple trials, with results averaged for statistical reliability.

**Methods And Evaluation Criteria:**

The paper introduces a latent exploration score to constrain the acquisition function optimization in a VAE's latent space. This constraint is designed as a continuous function of z, where higher values indicate greater likelihood that z lies within valid latent space regions.

**Other Comments Or Suggestions:**

On Page 4, Line 213, the term should be corrected to "quadratic function".

**Other Strengths And Weaknesses:**

While the proposed method presents an interesting approach with potential for scientific discovery, the experimental results currently lack sufficient empirical rigor to fully support its claims. Additionally, Theorem 3.1 appears to function more as a formulation or assumption rather than a demonstrably proven theorem. Due to these limitations, the methodological validity remains unconvincing in its current form.

**Questions For Authors:**

The paper would benefit from (1) a more explicit articulation of the specific challenges being addressed, and (2) a systematic demonstration of comparative limitations (e.g., failure cases of existing methods) to better contextualize its contributions. The discussion could be enhanced by explicitly revisiting Examples 1.1–1.3 in the conclusion, systematically comparing the proposed method’s generated expressions with those of baseline approaches. This would reinforce how the technique addresses the specific limitations identified earlier, providing a cohesive demonstration of its advancements.

**Relation To Broader Scientific Literature:**

The method could potentially be valuable for drug discovery applications.

**Theoretical Claims:**

The authors establish a theoretical framework representing Deep Generative Networks (DGNs) as Continuous Piecewise Affine (CPA) spline operators. Based on this formulation, they derive theorems characterizing the density function.

---

> ### Author Rebuttal · Authors · 2025-04-01
>
> We thank the reviewer for their time and effort in reviewing our work.
>
> ### Typos
>
> Thank you for pointing this out—we will update the text to replace "quadratic formula" with "quadratic function."
>
> ### Additional Visualizations
>
> We thank the reviewer for their great suggestion to add more visualizations to our manuscript.
> As Figure 1 provides a visualization of the dynamics within the latent space, we would be happy to include additional figures using different starting points in the revised version.
>
> ### Empirical Rigor
>
> Our main claim is that LES can be used as a robust method (across VAEs) to improve the number of valid solutions discovered through latent space optimization. We consider a rigorous empirical evaluation of this claim to be one that explores multiple VAE models, multiple notions of validity, and multiple latent space optimization tasks. This definition aligns with the goal of LSO, where we aim to use a VAE to find realistic solutions for a given objective.
>
> By this definition, we believe that our empirical evaluation is rigorous for the following reasons:
>
> - We study **22 VAEs**, varying both the decoder architecture and the beta parameters that control the trade-off between the prior and the reconstruction. This goes beyond previous related work, which studied only 4 VAEs [1].
> - We study **three notions of validity**, ensuring our empirical results are not specific to a single definition.
> - We evaluate **five different latent space optimization tasks**, all considered standard benchmarks in this literature [1,2].
>
> **References**
> [1] Notin, P., Hernández-Lobato, J.M., & Gal, Y. (2021). Improving black-box optimization in VAE latent space using decoder uncertainty. *Advances in Neural Information Processing Systems*, 34, 802–814.
> [2] Maus, N., Jones, H., Moore, J., Kusner, M.J., Bradshaw, J., & Gardner, J. (2022). Local latent space Bayesian optimization over structured inputs. *Advances in Neural Information Processing Systems*, 35, 34505–34518.
>
> ### Methodological Validity
>
> We believe that the validity of **Theorem 3.1** in our setup is well justified through the approximation results we cite. In addition, our extensive empirical evaluation further supports the soundness of our methodology.
>
> ### Failure Cases of Other Methods
>
> The failure mode we aim to address is that existing baselines tend to generate unrealistic solutions [1]:
>
> > “It has been widely observed that optimizing outside of the feasible region tends to give poor results, yielding samples that are low-quality, or even invalid (e.g., invalid molecular strings, non-grammatical sentences).”
>
> **Table 13** systematically demonstrates this widely known failure mode by comparing the percentage of valid solutions generated by each optimization method. In short, LES produces on average **62% valid solutions** more than any unregularized method—L-BFGS (26%), TurBO (38%), and GA (55%).
>
> **Reference**
> [1] Tripp, A., Daxberger, E., & Hernandez-Lobato, J.M.
> *Sample-efficient optimization in the latent space of deep generative models via weighted retraining.*
>
> ### Discussion Section
>
> We thank the reviewer for the valid suggestion to revisit Examples 1.1–1.3 in the discussion and to explicitly articulate the challenges being addressed, which we will happily do. A revised version of the first paragraph is included below:
>
> > “We propose LES to address over-exploration in latent space optimization (LSO). Desirable properties of LSO include stability—across tasks and VAE architectures—and the ability to consistently discover high-quality solutions that optimize a given objective. However, existing approaches often suffer from instability and excessive exploration, which can lead to unrealistic or invalid outputs without careful regularization. LES mitigates these issues by encouraging optimization to remain near the data manifold, thereby stabilizing LSO and improving the realism of generated solutions. LES is differentiable and fully parallelizable. Extensive evaluations demonstrate that incorporating LES as a penalty in LSO consistently enhances solution quality and objective outcomes. Moreover, LES outperforms alternative regularization techniques, proving to be the most robust across diverse datasets and varying definitions of validity—including syntactically valid expressions, chemically valid SMILES, and molecules that pass domain-specific property filters. In addition, LES has only a single hyperparameter (the regularization strength), and we observe empirically that deploying LES can provide significant performance gains. We therefore believe LES offers a powerful approach for discovering more realistic solutions, especially when the criteria for realism are difficult to define or validate.”

---

> > ### Comment · Reviewer_BNj6 · 2025-04-07
> >
> > Thank you to the author for addressing the rebuttal. After carefully reviewing the other reviewers' comments and revisiting the paper multiple times, I find the proposed idea interesting. However, I still feel that the paper lacks sufficient justification to fully support its claims.
> >
> > 1. The paper's structure and clarity could be improved. For example, the background section is challenging to follow—particularly the claim on line 113 that Gaussian processes are the most common surrogate model, which lacks a supporting reference. Moreover, the relevance of introducing Gaussian processes here is unclear, as it does not seem well-integrated into the broader discussion.
> >
> > 2. Regarding line 206, the claim that 'the prior is negligible' requires justification. Since this assumption plays a critical role in the theorem’s derivation, its validity should be properly established. Specifically: (1) under what conditions is this approximation valid, and (2) what quantitative bounds exist for the approximation error? The current presentation omits this essential reasoning without explanation, which significantly impacts the theorem’s rigor.
> >
> > 3. Additionally, since LES appears to bias the gradient using a regularization term, it would be helpful to include the original gradient descent direction in Figure 1 for comparison. The current figure, which spans 10 steps, makes it difficult to discern how the regularization term systematically biases the gradient at each step. A clearer visualization of this effect would strengthen the paper's exposition. Thus the motivation remains unclear to me.
> >
> > 4. The mathematical notation used in the paper is somewhat unconventional and unclear to me. More significantly, Theorem 3.1 does not appear to be a proper theorem in the conventional sense - it reads more like a proposal or definition. A proper theorem should be derived from clearly stated lemmas and assumptions to establish a rigorous conclusion, whereas the current formulation seems to simply present an algorithmic procedure for obtaining the regularization term. This makes the theoretical contribution and mathematical rigor difficult to assess.
> >
> > 5. The experiment section could be improved in several ways: First, proper citations are needed for the SMILES/SELFIES datasets introduced in line 232 and for the results shown in Table 1. Second, while 22 experiments were conducted, there's no ablation study analyzing critical factors like the impact of $\lambda$ - which likely affects performance differently across datasets. This omission leaves open questions about how the method's effectiveness varies under different conditions. Currently, the results are presented without clear progression or meaningful interpretation, making it difficult to draw substantive conclusions from the experimental data.
> >
> > Minor Type:
> > On line 233, should it be "\\;" but not ";" in latex instead?

---

> > > ### Author Response · Authors · 2025-04-08
> > >
> > > We thank the reviewer for their detailed response and address their comments below:
> > >
> > > 1. **Clarity of background and role of GPs:**
> > >    The reviewer makes a good point that while Gaussian Processes (GPs) are well accepted within both the Bayesian Optimization and latent space optimization communities, the clarity of the background could be improved. We will add [1] as a citation and include the following clarification in the background section: GPs are the standard choice in these domains due to their ability to model the full predictive distribution, which acquisition functions rely on. This addition will help contextualize our methodological choices more clearly.
> > >
> > > 2. **Justification for prior/log-det term balance:**
> > >    We provide below the average ratio between the prior term and the log-determinant term over a random sample of 500 data points from the training set to justify our claim that the prior terms is much smaller than the log-determinant term.
> > >
> > > Expressions
> > > | Architecture | Beta | Value  |
> > > |--------------|------|--------|
> > > | GRU          | 0.05 | 0.019  |
> > > | GRU          | 0.1  | 0.007  |
> > > | GRU          | 1    | 0.003  |
> > > | LSTM         | 0.05 | 0.039  |
> > > | LSTM         | 0.1  | 0.023  |
> > > | LSTM         | 1    | 0.003  |
> > > | Transformer  | 0.05 | 0.051  |
> > > | Transformer  | 0.1  | 0.033  |
> > > | Transformer  | 1    | 0.004  |
> > >
> > > SMILES
> > >
> > > | Architecture | Beta | Value   |
> > > |--------------|------|---------|
> > > | GRU          | 0.05 | 0.00027 |
> > > | GRU          | 0.1  | 0.00017 |
> > > | GRU          | 1    | 0.00003 |
> > > | LSTM         | 0.05 | 0.00028 |
> > > | LSTM         | 0.1  | 0.00015 |
> > > | LSTM         | 1    | 0.00004 |
> > > | Transformer  | 0.05 | 0.00026 |
> > > | Transformer  | 0.1  | 0.00015 |
> > > | Transformer  | 1    | 0.00004 |
> > >
> > > SELFIES
> > > | Architecture | Beta | Value   |
> > > |--------------|------|---------|
> > > | Transformer  | 0.05 | 0.00024 |
> > > | Transformer  | 0.1  | 0.00015 |
> > > | Transformer  | 1    | 0.00003 |
> > > | Transformer (maus et al) | -| 0.00004|
> > >
> > > 3. **Unregularized trajectory visualization:**
> > >    We agree that it is important to display the original (unregularized) trajectory. Indeed, as described in the figure caption: *"Optimization trajectories with (blue) and without (red) LES constraint in the latent space of a VAE."* The red trajectory ends at an invalid solution, motivating the need for LES. As we explain: *"High LES values correlate with valid areas, and incorporating LES in LSO produces an expression that adheres to the grammatical rules of Example 1.1."*
> > >
> > > 4. **Theorem 3.1 rigor and assumptions:**
> > > Theorem 3.1 is indeed a fully rigorous result, derived under the explicitly stated assumptions (line 180): "Assume that $L_\theta$ is bijective and can be expressed as a CPA." We also provide justification for these assumptions in our discussion of the theorem's limitations. Specifically, the CPA assumption is supported by a function approximation argument (lines 220–227), while the bijectivity assumption is motivated by the observation that decoders which are not bijective (almost everywhere) tend to exhibit degeneracies that are rarely encountered in practice (lines 228–241).
> > >
> > > 5. **Ablation on the effect of $\lambda$:**
> > >    The reviewer is correct to point out, as Reviewer SqeL did, that our original submission did not include a study on the effect of $\lambda$. In our response to Reviewer SqeL, we provided such a study, and this will be included in the camera-ready version of our paper.
> > >
> > > Thanks for catching the typo, which will be corrected!
> > >
> > > ---
> > >
> > > [1] Frazier, P.I., 2018. A tutorial on Bayesian optimization. *arXiv preprint arXiv:1807.02811*.

---

### Official Review · Reviewer_SqeL · 2025-03-21

**Overall Recommendation:** 3

**Summary:**

The paper proposes LES, a novel method for mitigating the over-exploration phenomenon in latent space optimization for discrete black-box optimization. LES can be easily integrated as a regularizer for acquisition function optimization. Experiment results validate that LES can improve the performance of latent space optimization methods.

**Claims And Evidence:**

The authors present their claim with clear and convincing evidence.

**Essential References Not Discussed:**

N/A

**Experimental Designs Or Analyses:**

The authors follow (as far as I know) the conventional procedure for latent-space Bayesian optimization.

**Methods And Evaluation Criteria:**

The authors follow (as far as I know) the conventional procedure for latent-space Bayesian optimization.

**Other Comments Or Suggestions:**

Here are some questions:

- Authors use TuRBO as a baseline. When I read the experiment setup part, the authors used logEI as a base acquisition function. Do authors also apply LogEI for TuRBO as stated in [1]? For a fair comparison, it seems that all baselines use the same acquisition function.

- For Table 10, it seems that regularizing the acquisition function with likelihood score is the second-best baseline. Could authors also provide ablation studies on $\lambda$ for the likelihood baseline? I think it is crucial, as the likelihood score also has a single hyperparameter and is faster to compute. If we can achieve a similar performance by tuning the hyperparameter for the likelihood score, the significance of the proposed method greatly decreases. I hope the authors verify this part carefully.

[1] Ament, Sebastian, et al. "Unexpected improvements to expected improvement for bayesian optimization." Advances in Neural Information Processing Systems 36 (2023): 20577-20612.

**Other Strengths And Weaknesses:**

**Strengths**

- As a result, LES can be implemented as a regularizer when we optimize the acquisition function during the Bayesian optimization procedure, which is straightforward.
- The authors try to validate that LES can mitigate the over-exploration problem by toy experiment setting and try to validate that LES does not suffer from high time-complexity.

**Weakness**

- According to the experiment setup, the authors employ a deep kernel to reduce the dimension of latent space into 12 dimensions, which is quite low compared to the original setup of [1]. Is there any reason why such kind of dimension reduction is required? Otherwise, is it hard to achieve improvement using LES when the latent dimension is high?

- I find that there are several ablation studies on hyperparameters of baselines. However, it is hard to find the ablation of hyperparameters such as $\lambda$ in terms of performance. While I find that using LES can lead to a high proportion of valid solutions, I wonder if the suggested method is robust to different hyperparameter configurations in terms of performance.

[1] Maus, Natalie, et al. "Local latent space bayesian optimization over structured inputs." Advances in neural information processing systems 35 (2022): 34505-34518.

**Questions For Authors:**

Please see above.

**Relation To Broader Scientific Literature:**

Discrete black-box optimization can be used for scientific discovery problems.

**Theoretical Claims:**

I checked that the Theorem 3.1 is valid.

---

> ### Author Rebuttal · Authors · 2025-04-01
>
> We thank the reviewer for their time and effort in reviewing our work.
>
> **Deep Kernel**
>
> The deep kernel affects only the computation of the kernel in the GP surrogate model—it does not reduce the dimensionality of the latent space on which LES is applied. Based on our results with the SELFIES VAE, we see no indication that achieving improvement becomes more difficult in higher-dimensional settings.
>
> **TuRBO Method**
>
> The TuRBO method (similar to every other method in this work) uses the LogEI acquisition function, for the same reasons noted by the reviewer. We will add clarification in our revised version, in section 4 under LSO setup paragraph.
>
> **Ablation studies on $\lambda$**
>
> As requested by the reviewer, we provide an ablation study of the parameter $\lambda$ for both LES and the Likelihood score. Due to space constraints, we present the full results for the average of the top 20 solutions—since this metric tends to exhibit lower variance—and summarize the key findings for both the best and top-20 solutions below.
> For the average of the top 20 solutions, we find that using a lower $\lambda$ (0.1) for the likelihood score significantly improves performance on the RANO task when using the pre-trained SELFIES VAE, increasing the score from 0.31 to 0.41. We will include this setting in both the appendix and the main text of the revised version.
> For the best solution, we observe the following notable differences:
> For the expressions transformer with $\beta = 1$, increasing the regularization to $\lambda = 0.2$ improves both LES and the likelihood scores to -0.46 and -0.42, respectively.
>
>
> For the SMILES transformer with $\beta = 0.1$, setting $\lambda = 0.8$ improves the likelihood from 3.16 to 3.25, surpassing the 3.23 achieved by LES at $\lambda = 0.5$.
>
> *Expressions*
>
> **LES**
>
> | Architecture | Beta |0.05 |0.1 |0.2 |
> |--|--|----------------|--|--|
> | GRU| 0.05 | -1.43| -1.28 | -1.32 |
> | GRU| 0.1| -1.34| -1.06 | -1.29 |
> | GRU| 1| -0.84| -0.98 | -1.08 |
> | LSTM | 0.05 | -1.06| -1.09 | -1.00 |
> | LSTM | 0.1| -0.80| -0.72 | -0.73 |
> | LSTM | 1| -1.79| -1.79 | -1.84 |
> | Transformer| 0.05 | -1.00| -0.86 | -0.92 |
> | Transformer| 0.1| -0.77| -0.75 | -0.67 |
> | Transformer| 1| -1.36| -1.28 | -1.05 |
>
> **Likelihood**
>
> | Arch | Beta |0.05 |0.1 |0.2 |
> |--|--|----------------|--|--|
> | GRU| 0.05 | -1.15| -1.10 | -1.11 |
> | GRU| 0.1| -1.31| -1.16 | -1.14 |
> | GRU| 1| -0.88| -1.04 | -1.15 |
> | LSTM | 0.05 | -1.00| -1.05 | -1.17 |
> | LSTM | 0.1| -0.75| -0.81 | -0.72 |
> | LSTM | 1| -1.81| -1.79 | -1.78 |
> | Transformer| 0.05 | -0.93| -0.95 | -0.98 |
> | Transformer| 0.1| -0.81| -0.74 | -0.70 |
> | Transformer| 1| -1.45| -1.42 | -1.19 |
>
> *SMILES*
>
>
> **LES**
>
> | Architecture | Beta |0.3 |0.5 |0.8 |
> |--|--|--|--|--|
> | GRU| 0.05 | 2.20| 2.31| 2.32|
> | GRU| 0.1| 2.12| 2.30| 2.35|
> | GRU| 1| 0.38| 1.64| 1.77|
> | LSTM | 0.05 | 2.34| 2.33| 2.37|
> | LSTM | 0.1| 0.94| 1.57| 1.80|
> | LSTM | 1| 0.59| 1.94| 2.17|
> | Transformer| 0.05 | 2.30| 2.26| 2.35|
> | Transformer| 0.1| 2.31| 2.26| 2.29|
> | Transformer| 1| 2.09| 2.17| 2.30|
>
> **Likelihood**
>
> | Architecture | Beta |0.3 |0.5 |0.8 |
> |--|--|--|--|--|
> | GRU| 0.05 | 2.15| 2.31| 2.25|
> | GRU| 0.1| 2.10| 2.12| 2.22|
> | GRU| 1| 0.60| 1.49| 1.40|
> | LSTM | 0.05 | 2.17| 2.28| 2.30|
> | LSTM | 0.1| 0.81| 1.43| 1.58|
> | LSTM | 1| 0.53| 1.67| 1.77|
> | Transformer| 0.05 | 2.43| 2.25| 2.32|
> | Transformer| 0.1| 2.26| 2.23| 2.34|
> | Transformer| 1| 2.01| 2.16| 1.98|
>
> *SELFIES*
>
>
> **LES**
>
> | Objective | Beta |0.1 |0.3 |0.5 |
> |--|--|--|--|--|
> | Pdop| 0.05 | 0.37| 0.37| 0.37|
> | Pdop| 0.1| 0.36| 0.36| 0.36|
> | Pdop| 1| 0.35| 0.32| 0.33|
> | Rano| 0.05 | 0.21| 0.22| 0.21|
> | Rano| 0.1| 0.21| 0.21| 0.19|
> | Rano| 1| 0.21| 0.21| 0.20|
> | Zale| 0.05 | 0.33| 0.34| 0.34|
> | Zale| 0.1| 0.34| 0.33| 0.33|
> | Zale| 1| 0.31| 0.31| 0.31|
>
> **Likelihood**
>
> | Objective | Beta |0.1 |0.3 |0.5 |
> |--|--|--|--|--|
> | Pdop| 0.05 | 0.37| 0.37| 0.37|
> | Pdop| 0.1| 0.36| 0.36| 0.34|
> | Pdop| 1| 0.34| 0.29| 0.28|
> | Rano| 0.05 | 0.21| 0.22| 0.21|
> | Rano| 0.1| 0.22| 0.21| 0.20|
> | Rano| 1| 0.21| 0.23| 0.19|
> | Zale| 0.05 | 0.33| 0.33| 0.32|
> | Zale| 0.1| 0.34| 0.33| 0.32|
> | Zale| 1| 0.30| 0.29| 0.26|
>
> **SELFIES (Maus et al)**
>
> **LES**
>
> | Objective | Beta | 0.1 | 0.3 | 0.5 |
> |-----------|------|---------------|---------------|---------------|
> | Pdop      | –    | 0.46          | 0.48          | 0.47          |
> | Rano      | –    | 0.30          | 0.29          | 0.30          |
> | Zale      | –    | 0.40          | 0.40          | 0.41          |
>
> **Likelihood**
>
> | Objective | Beta | Lambda = 0.1 | Lambda = 0.3 | Lambda = 0.5 |
> |-----------|------|---------------|---------------|---------------|
> | Pdop      | –    | 0.42          | 0.40          | 0.35          |
> | Rano      | –    | 0.22          | 0.29          | 0.26          |
> | Zale      | –    | 0.41          | 0.39          | 0.31          |

---

> > ### Comment · Reviewer_SqeL · 2025-04-07
> >
> > Sorry, I sent an official comment and just found that it is not visible to the authors...
> >
> > Thank you for your clear explanation and additional experiments. I hope additional experiments will be included in the final manuscript. I just adjusted the score.

---

### Official Review · Reviewer_D91d · 2025-03-24

**Overall Recommendation:** 4

**Summary:**

This paper addresses the problem of latent space optimization (LSO) for combinatorial problems. LSO uses an encoder-decoder architecture to map from a continuous latent space to a combinatorial space. The optimization can be carried out in a continuous space which has the advantage that it allows for continuous optimization.

When sampling from the latent space, one often encounters invalid solutions (for instance, invalid arithmetic expressions if the problem at hand is to optimize such).

The authors propose a score (latent exploration score - LES) that promotes valid solutions and can be incorporated into the optimization objective as a regularization term. The authors show that doing so improves performance. They further validate that their score achieves higher density for valid solutions, which they use as a proxy for LES to avoid over-exploration.

**Claims And Evidence:**

The main claims of this paper are that 1) a high LES is likely to yield a valid point and 2) that incorporating LES in the black-box optimization improves performance.

Claim 1 is mainly supported by Table 2 (see my questions below regarding this table), which compares LES to other scores (or ‘no score’ = ‘Prior’). LES improves the likelihood of a valid point and is better than the other methods (Uncertainty and Likelihood).

Claim 2 is supported by Figure 3, which shows that LES improves performance or is competitive with the state of the art.

Other claims:

- Recurrent neural networks such as LSTMs or GRU networks can be approximated with high accuracy os PCA functions so that the theoretical results apply. This is supported by references but not shown explicitly in the paper.
- LES gives high values for valid points in the latent space. This is supported by empirical evidence.

**Essential References Not Discussed:**

The paper covers the essential references related to its contributions. One paper that should be added is “Tripp, Austin, Erik Daxberger, and José Miguel Hernández-Lobato. ‘Sample-efficient optimization in the latent space of deep generative models via weighted retraining.’ Advances in Neural Information Processing Systems 33 (2020): 11259-11272."

**Experimental Designs Or Analyses:**

The experimental designs are comprehensive. The authors conduct a thorough analysis across multiple datasets and VAE architectures, providing a robust evaluation of LES.

**Methods And Evaluation Criteria:**

The evaluation criteria and benchmark datasets are chosen reasonably.

**Other Comments Or Suggestions:**

There is a broken sentence in the second paragraph of ‘Over-exploration in LSO’ starting with ‘While acquisition functions…’

The main problem should be introduced with an additional sentence in the abstract.

The authors sometimes refer to LES as a ‘constraint,’ but it’s not really a constraint. I suggest only using the term ‘score’.

**Other Strengths And Weaknesses:**

Table 2 is confusing. $\beta$ has not been introduced before. It is mentioned later in the text, but it does not refer to that table again. The overall section around this table should be revised. I would expect the training data, VAE prior, and out-of-distribution samples to be shown separately in the table, but that doesn’t seem to be the case. It almost seems like something is missing, or I’m misunderstanding something, so the text should be revised.

The motivation for the definition of LES is very brief. Why is it chosen as the logarithm of the determinant term? What is the intuition behind this choice?

**Questions For Authors:**

Below Eq. (5), the authors refer to results from Daubechies et al. What do those say?

Right above Eq. (1), what exactly is the reason for the empty category?

Right above ‘Optimization tasks’ in the right column of page 6: What exactly do you mean when you say the acquisition function is sequentially maximized?

**Relation To Broader Scientific Literature:**

The LSE is particularly relevant for latent-space black-box optimization problems where the decoder can generate invalid solutions. The domain of problems this method can be applied to is relatively broad and the problem is quite relevant.

**Theoretical Claims:**

I did not check the correctness of the proofs.

---

> ### Author Rebuttal · Authors · 2025-04-01
>
> We thank the reviewer for their time and effort in reviewing our work.
>
> We address the reviewer's questions and comments chronologically.
>
> Thank you for suggesting the addition of the important work by Tripp et al. We would like to point out that we refer to this work in line 155:
>
> > “The frequent generation of invalid solutions during acquisition optimization, which implies that the estimated uncertainty can be problematic in this setting, underscores the need for additional regularization (Tripp et al., 2020), which we aim to address.”
>
> We thank the reviewer for their suggestion to clarify the meaning of the parameter **beta**, which balances the alignment with the prior and the reconstruction loss. We will add a description of this parameter in the first paragraph of Section 3.2.
>
> Due to margin constraints, **Table 2** is a truncated version of the full **Table 11**, which is deferred to the appendix. In the revised version, we will make sure to clarify this point in the last paragraph of Section 3.2.
>
> The intuition behind using the log-determinant terms is explained in lines 172–188. In short:
>
> > “Why use the sequence density function? We argue that for a well-trained decoder network, the density should be higher in areas of the sequence space close to the training data.”
>
> As we mention in line 206:
>
> > “...as the contribution of the prior is negligible in magnitude in all the decoders we study,”
>
> this is why we only use the log-determinant term.
>
> We will make sure to be consistent in using the term **score** when referring to LES—thank you for the suggestion.
>
> The results from Daubechies state that ReLU networks (which are CPA spline functions) can approximate refinable functions—a class of non-smooth functions—in addition to known results on other classes of functions (e.g., Weierstrass, polynomials, and analytic functions), as described in this work. This clarification will be added to the limitations of Theorem 3.1.
>
> The **EMPTY** category allows generating sequences shorter than the maximal length. For example, if the maximal length is 3, then we can generate the expression `"x"` as `"x<EMPTY><EMPTY>"`. This example will be added to the first paragraph of Section 2 to enhance its clarity.
>
> **Sequential optimization** means that we generate each solution separately, rather than optimizing all the solutions jointly. This setup was used in the work by Notin et al., which we follow. This clarification will be added to the LSO setup paragraph in Section 4.

---

> > ### Comment · Reviewer_D91d · 2025-04-01
> >
> > Thank you for your answers, and I'm sorry for missing the Tripp reference. I think this paper has a strong contribution and mainly suffers from minor layout or typo issues that can be addressed for the camera-ready version. I adjusted my score.

---

### Official Review · Reviewer_eqXz · 2025-03-25

**Overall Recommendation:** 3

**Summary:**

The authors propose a scoring method, Latent Exploration Score (LES), for Latent Space Optimization (LSO), to mitigate overexploration, which often results in unrealistic solutions in discrete optimization problems. LES uses the VAE decoder’s data distribution without additional training or architectural changes, incorporating molecule density in the latent space. The empirical results show that LES performs better or similar most frequently across 30 tasks compared to baselines.

**Claims And Evidence:**

The authors make several key claims about their method most of which are supported by empirical evidence. Authors, in particular, claim the LES’ robustness in generating valid solutions and achieving high objective values. However, isn’t LES still closely linked to the decoder’s performance? Even if the latent space is optimized, if the decoder struggles with out-of-distribution latent points, will this also impact LES’s effectiveness?

**Essential References Not Discussed:**

The paper provides sufficient relevant background and work to understand the context of the paper.

**Experimental Designs Or Analyses:**

The experimental design includes extensive comparisons across various datasets and methods. However, the authors could improve the discussion by addressing potential failure cases of LES, particularly for datasets or tasks involving out-of-distribution latent spaces. Additionally, the number of hyperparameters differs from those used in the benchmark methods. Since a grid search was conducted for these hyperparameters, resulting in an unequal compute budget, I think it is a bit unfair to compare LES with the baselines, as this could potentially lead to misleading performance conclusions.

**Methods And Evaluation Criteria:**

The proposed methods and evaluation criteria are mostly well-written and make sense for the LSO problem. The use of top-1 and top-20 averages provides several insights into the performance.

**Other Comments Or Suggestions:**

Minor typos:
* Line 230: What is ScaLES? I think it’s not defined in the paper.
* Line 226: Shouldn’t it be Theorem 3.1?

**Other Strengths And Weaknesses:**

* The use of decoder likelihood as a latent validity heuristic is well-motivated.
* The paper provides detailed derivations and adequate explanations.
* The method is more efficient than some baseline approaches, though this advantage may diminish for higher-dimensional latent spaces.

**Questions For Authors:**

Please refer to my previous comments.

**Relation To Broader Scientific Literature:**

The paper refers to some important previous work and other baselines in latent space optimization (LSO). The authors explain how LES builds on and improves over baselines, especially addressing robustness across benchmarks and tasks.

**Theoretical Claims:**

The paper includes several derivations, including Theorem 3.1, which derives a likelihood-based formula for evaluating latent validity. The derivation appears correct.

---

> ### Author Rebuttal · Authors · 2025-04-01
>
> We thank the reviewer for their time and effort in reviewing our work, and we are particularly glad to see that the reviewer agrees that our claims regarding the effectiveness of LES are supported by our empirical evidence, which the reviewer described as “extensive”.
>
> First, thank you for catching our typos, which have been corrected (`ScaLES → LES`, `Theorem 5 → Theorem 3.1`).
>
> The reviewer is correct to point out that LES is closely related to decoder performance. As this is an important point, we made sure to include it under the limitations of Theorem 3.1 (lines 247–252):
>
> > “Lastly, it is crucial to highlight that the ability of LES to detect out-of-distribution data is closely tied to the decoder’s capacity to accurately model the data. Although Theorem 3.1 holds for poorly trained decoders under the given conditions, we advise against relying on LES in such cases.”
>
> Indeed, we expect the decoder to struggle with OOD data points in general (this has also been widely observed in many previous works that we cite [1]):
>
> > “...therefore all LSO methods known to us employ some sort of measure to restrict the optimization to near or within the feasible region.”
>
> Lastly, we would like to offer some clarification regarding **hyperparameter tuning** (an important clarification which will be added to our submission).
>
> We acknowledge that some manual selection of the hyperparameter $\lambda$ was performed based on the level of regularization needed for each task, which could give our method a slight advantage over the baseline. To ensure our improvements are not merely the result of this tuning, we included an ablation study for **TurBO** and **L-BFGS** in our original submission, which demonstrates that LES would out-perform these methods on average even if the best hyperparameter was retroactively selected.
>
> We also include an additional ablation and analysis of $\lambda$ selection in our response to Reviewer SqeL.
>
> Given this, we believe our comparisons are fair and that our conclusions are well supported: LES offers a consistent and meaningful improvement over the baselines, and these gains are not artifacts of hyperparameter tuning.
>
> **Reference**
>
> [1] Tripp, A., Daxberger, E., & Hernandez-Lobato, J.M.
> *Sample-efficient optimization in the latent space of deep generative models via weighted retraining.*

---

### Decision · Program_Chairs · 2025-05-01

**Decision:**

Accept (poster)

**Comment:**

To address the issue of over-exploration in existing Latent Space Optimization (LSO) methods, the authors introduce the Latent Exploration Score (LES). LES serves as a constraint within the LSO framework to guide the optimization process more effectively. Formally, it is defined as the output log-density of the decoder, helping to regulate the search in latent space and prevent overly aggressive exploration.

Almost all reviewers agree that the paper merits acceptance, and after carefully reading the rebuttal and discussion, I tend to concur. Please incorporate the reviewers’ feedback into the final version of the paper, with particular attention to the following concerns:

1- Incorporate the hyper-parameter tuning experiments

2- Conduct additional ablation studies for the extra factors and parameters

3- Provide further clarification regarding the comparisons with certain baselines